# Significance of Programmed Cell Death Pathways in Neurodegenerative Diseases

**DOI:** 10.3390/ijms25189947

**Published:** 2024-09-15

**Authors:** Dong Guo, Zhihao Liu, Jinglin Zhou, Chongrong Ke, Daliang Li

**Affiliations:** 1College of Life Science, Fujian Normal University Qishan Campus, Fuzhou 350117, China; qsx20221403@student.fjnu.edu.cn (D.G.); qsx20221400@student.fjnu.edu.cn (Z.L.); qsx20211380@student.fjnu.edu.cn (J.Z.); 2Fujian Key Laboratory of Innate Immune Biology, Biomedical Research Center of South China, Fujian Normal University Qishan Campus, Fuzhou 350117, China

**Keywords:** programmed cell death, neurodegenerative diseases (NDDs), apoptosis

## Abstract

Programmed cell death (PCD) is a form of cell death distinct from accidental cell death (ACD) and is also referred to as regulated cell death (RCD). Typically, PCD signaling events are precisely regulated by various biomolecules in both spatial and temporal contexts to promote neuronal development, establish neural architecture, and shape the central nervous system (CNS), although the role of PCD extends beyond the CNS. Abnormalities in PCD signaling cascades contribute to the irreversible loss of neuronal cells and function, leading to the onset and progression of neurodegenerative diseases. In this review, we summarize the molecular processes and features of different modalities of PCD, including apoptosis, necroptosis, pyroptosis, ferroptosis, cuproptosis, and other novel forms of PCD, and their effects on the pathogenesis of neurodegenerative diseases, such as Alzheimer’s disease (AD), Parkinson’s disease (PD), Huntington’s disease (HD), amyotrophic lateral sclerosis (ALS), spinal muscular atrophy (SMA), multiple sclerosis (MS), traumatic brain injury (TBI), and stroke. Additionally, we examine the key factors involved in these PCD signaling pathways and discuss the potential for their development as therapeutic targets and strategies. Therefore, therapeutic strategies targeting the inhibition or facilitation of PCD signaling pathways offer a promising approach for clinical applications in treating neurodegenerative diseases.

## 1. Introduction

The Nomenclature of Cell Death Committee has established guidelines dividing cell death into two distinct categories: accidental cell death (ACD) and programmed cell death (PCD) [1]. ACD is an uncontrolled cellular process that occurs in response to accidental injury stimuli, such as necrosis [2]. PCD is essential for maintaining physiological homeostasis in mammals by clearing damaged cells, facilitating tissue renewal, and supporting organismal development, all of which are strictly regulated by intracellular signaling cascades [3]. The hallmark ultrastructural features of cells undergoing PCD, including cytoplasmic shrinkage, nuclear condensation, and chromatin fragmentation, were first observed in 1972, leading to the definition of this form of PCD as apoptosis. These features are evident in various tissues under both physiological and certain pathological conditions [4]. In the past five decades, novel forms of PCD and their corresponding signaling cascades have been identified, including necroptosis, pyroptosis, ferroptosis, cuproptosis, mitochondrial permeability transition (MPT)-driven necrosis, autophagy-dependent cell death (ADCD), lysosome-dependent cell death (LDCD), parthanatos, alkaliptosis, oxeiptosis, NET-release-induced necrotic cell death (NETosis), entotic cell death (ENTosis), and disulfidptosis [5,6,7].

The etiology of neurodegenerative diseases (NDDs) is multifactorial and is associated with abnormalities in various intracellular processes, such as autophagy, mitochondrial biogenesis, homeostasis of the endoplasmic reticulum (ER), and epigenetic modifications [8]. The most well-known NDDs include Alzheimer’s disease (AD), Parkinson’s disease (PD), Huntington’s disease (HD), amyotrophic lateral sclerosis (ALS), spinal muscular atrophy (SMA), multiple sclerosis (MS), traumatic brain injury (TBI), and stroke. As the global population continues to grow and age, NDDs have become one of the foremost medical and social concerns worldwide. According to clinical data, the number of people suffering from Parkinson’s disease (PD) tripled between 1996 and 2016 [9]. The prevalence of AD in China among those aged over 60 was 1.37%, and more than 50 million people are affected by AD worldwide [10]. The incidence of HD, ALS, SMA, MS, TBI, and stroke has also been increasing annually worldwide [11,12,13,14,15]. Notably, stroke remains the second-leading cause of death worldwide, with the estimated global cost of stroke being approximately 0.66% of global GDP [16].

The central nervous system (CNS) comprises the brain and spinal cord. Under normal conditions, PCD signaling cascades are tightly regulated at temporal and spatial levels to establish neural architecture in the CNS [17]. During normal neural embryonic and postnatal development, apoptosis controls the survival of embryonic stem cells that have the appropriate size and shape and have made proper connections with their axons and neurites [18]. In addition, regulators of apoptosis play a crucial role in cell survival during developmental neurogenesis, such as the anti-apoptotic Bcl-2 family members myeloid cell leukemia-1 (MCL-1) and Bcl-2-related gene long isoform (Bcl-XL) [19,20,21]. However, aberrant neuronal cell death is a hallmark of the pathology associated with NDDs, and different PCD pathways interact in the progression of these diseases [22]. In recent years, numerous clinical trials have focused on PCD pathways to develop therapeutic strategies for the treatment of NDDs, achieving inspiring progress.

In this review, we provide a comprehensive overview of the signaling pathways involved in various PCD subroutines. Subsequently, we elucidate the similarities and differences among these pathways. We further discuss the role of different PCD pathways in the pathogenesis and progression of NDDs. Finally, we discuss existing and potential therapeutic strategies focusing on the central regulators of various PCD pathways for the treatment of NDDs.

## 2. Forms of Programmed Cell Death

### 2.1. Apoptosis

There are two distinct pathways that trigger apoptosis: the intrinsic and extrinsic pathways. The intrinsic pathway, also called mitochondrial or Bcl-2-regulated apoptosis, is characterized by non-receptor-mediated initiation and mitochondrial dependence in response to intracellular stress, such as DNA damage, ER stress, hypoxia, extremely high concentrations of cytosolic calcium, microtubular alteration, and growth factor deprivation [23]. After the generation of intracellular stimuli, the expression of BH3 (bcl-2 homolog3r)-only proteins, including BIM, PUMA, BID, BMF, BAD, HRK, BIK, and NOXA, is upregulated. These BH3-only proteins then bind to anti-apoptotic proteins, such as Bcl-2, Bcl-XL, and Mcl-1, to liberate and activate pro-apoptotic proteins, such as BAX, BAK, and BOK [24,25,26]. Subsequently, pro-apoptotic proteins undergo oligomerization, causing the dissipation of mitochondrial membrane potential. This leads to the disruption of mitochondrial outer membrane permeability (MOMP) and the formation of the mitochondrial permeability transition pore (MPT), allowing apoptogenic factors such as cytochrome-c and small mitochondria-derived activator of caspase (Smac) to be released into the cytosol [27,28,29]. Of note, cytochrome-c binds to the apoptosis protease activating factor-1 (Apaf-1) and induces the formation of the apoptosome, which recruits procaspase-9 for its cleavage and activation. Next, activated caspase-9 cleaves and activates its downstream effectors, including caspase-3, caspase-6, and caspase-7 [30]. The presence of Smac in the cytosol prevents the activation of inhibitor of apoptosis proteins (IAPs) through direct binding, thereby allowing for the initiation of caspase-dependent pathways [31]. These apoptogenic factors aim to induce the activation of caspase-dependent cascades, resulting in the cleavage of hundreds of proteins and ultimately apoptosis. Similarly, ER stress triggered by an imbalance in calcium homeostasis can induce the expression of caspase-12, which is localized in the ER membrane, and then recruits caspase-7 to the ER membrane to initiate apoptosis [32,33,34]. In addition, apoptosis-inducing factors (AIFs) can induce apoptosis independently of caspase signals [35]. After the cleavage of calcium-dependent proteases, especially calpain, AIFs are translocated from the inner membrane of mitochondria to the nucleus by nuclear localization signals (NLSs), leading to genome instability and chromatin fragmentation [36].

The extrinsic pathway, also known as death receptor apoptosis, is induced by the interaction between extracellular ligands and death receptors anchored in the cell membrane [37]. The extracellular ligands consist mainly of the tumor necrosis factor (TNF) superfamily, Fas ligand (FasL), and TNF-related apoptosis-inducing ligand (TRAIL), which bind to death receptors, such as TNF receptor (TNFR)-1, TNFR-2, Fas, TRAILR1, and TRAILR2 [38]. Upon the binding of ligands and death receptors, the death receptors undergo oligomerization and conformational changes to expose their death domain (DD) for the recruitment of TNF receptor-associated death domain (TRADD) and Fas-associated death domain (FADD) adaptor proteins, leading to the formation of an intracellular death-inducing signaling complex (DISC) for Fas-FasL and TRAIL-TRAILR, as well as complex II for TNF-TNFR [39,40]. Next, DISC and complex II mediate the cleavage and activation of procaspase-8, initiating the cleavage of caspase-3 and caspase-7 to induce apoptotic signaling pathways and the proteolytic degradation of a variety of intracellular proteins [37,39,41]. In some specific situations, apoptosis cannot be activated by the extrinsic pathways. Therefore, it is necessary for cleaved caspase-8 to interact with BH3-interacting domain death agonist (BID) and then cleave BID to form activated tBID. tBID subsequently directly activates pro-apoptotic multi-domain proteins to induce MOMP and mitochondrial apoptosis [17,37,40]. The mechanisms of apoptotic pathways are depicted in Figure 1.

### 2.2. Necroptosis

Necroptosis can be regarded as a regulated form of necrosis, first introduced in 2005 [42]. Canonical necroptosis is an alternative to apoptosis because its activation relies on the engagement of apoptotic extracellular ligands and their corresponding death receptors when the activation of caspase-8 is inhibited by pharmacological agents or viral inhibitors [43,44]. After the binding of ligands and receptors, such as TNF-TNFR, FAS-FASL, and TRAIL-TRAILR, DD recruits receptor-interacting serine/threonine protein kinase 1 (RIPK1) through homotypic binding, and then RIPK1 undergoes autophosphorylation [45]. Phosphorylated RIPK1 binds and phosphorylates RIPK3 through shared RIP (receptor-interacting protein) homology interaction motifs (RHIMs) [46,47,48]. Subsequently, mixed-lineage kinase like (MLKL) is phosphorylated by the RIPK1-RIPK3 complex and then undergoes oligomerization to form a high-molecular-weight complex, termed the necrosome, in the cytosol [49,50]. Next, the necrosome translocates to the plasma membrane, causing rupture of the plasma membrane, cell swelling, the release of cytokines and chemokines, as well as potassium efflux, leading to inflammation and immune responses [51]. Similarly, tumor-cell-derived amyloid precursor protein (APP) can activate the RIPK1-RIPK3-MLKL axis-induced necroptosis through binding to death receptor 6 (DR6) in endothelial cells, which enhances the extravasation of circulating tumor cells (CTCs) [52,53]. In addition, viral RNA and DNA or RNA leaked from damaged mitochondria can also induce necroptosis by activating RHIM-containing Z-dsDNA/dsRNA-binding protein (ZBP1), which subsequently results in the activation of the RIPK3-MLKL axis [54,55]. Meanwhile, necroptosis can also be triggered by the recognition of toll-like receptor 3 (TLR3) and TLR4 to double-stranded RNA (dsRNA) from viruses and lipopolysaccharide (LPS) from bacteria, respectively [56,57]. Upon binding, TLR3 and TLR4 are capable of activating TIR-domain-containing adapter-inducing interferon-β (TRIF) containing RHIM, ensuing the activation of the RIPK3-MLKL axis [43]. Additionally, proinflammatory factors such as interferons (IFNs) can trigger RIPK1-RIPK3-MLKL axis-mediated necroptosis or ZBP1-RIPK3-MLKL axis-mediated necroptosis in the absence of RIPK1 via sensing IFN receptors (IFNRs), suggesting that necroptosis is essential for the induction of inflammation [58]. Figure 1 depicts the activation of various necroptotic pathways.

### 2.3. Pyroptosis

Pyroptosis, a proinflammatory form of programmed cell death first discovered in 1989, is triggered by a variety of inflammasomes and is executed by the caspase and gasdermin (GSDM) families [59]. Pyroptotic signaling cascades include the canonical, non-canonical, caspase-3-induced, caspase-8-induced, and granzyme (GZM)-mediated pathways [60]. The purpose of the canonical pathway is to respond to pathogen invasion and facilitate the development of adaptive immune responses [60]. Upon the activation of pattern recognition receptors (PRRs) by pathogenic exposures, such as bacteria and viruses, PRRs recruit pro-caspase-1 and apoptosis-associated speck-like protein containing a caspase recruitment domain (CARD) (ASC) to assemble the inflammasome. After that, pro-caspase-1 undergoes self-cleavage to form activated caspase-1, leading to the cleavage of GSDMD and the release of its cytotoxic N-terminal P30 fragment containing the pore formation domain (PFD) as well as the synthesis of IL-18 and IL-1β [61]. Finally, the N-terminus inserts into the plasma membrane and oligomerizes to form pores with inner diameters of approximately 12–14 nm in the plasma membrane, leading to the expulsion of proinflammatory factors, chromatin degradation, and cell swelling [62]. The non-canonical pathway is triggered by the infection of Gram-negative bacteria [63]. In mice, caspase-11, or in humans, caspase-4 and caspase-5, can be activated by bacterial LPS through CARD, after which they proteolytically hydrolyze GSDMD to release the N-terminus containing PFD. The N-terminus finally undergoes oligomerization to translocate to the plasma membrane and cause plasma membrane perforation [64]. Significantly, activated caspase-11, caspase-4, and caspase-5 can also be packaged into the NLRP3 inflammasome to induce pyroptosis through cleaving GSDMD and causing the efflux of potassium ions (K^+^) [65,66,67]. In addition, activated caspase-11 can also cause pyroptosis through activating the Pannexin-1-ATP-P2X7 channel and the efflux of potassium ions (K^+^) [68]. Caspase-4, caspase-5, and caspase-11 participate in the maturation and secretion of IL-18 and IL-1β rather than their synthesis [69]. The protein phosphatase PtpB from Mycobacterium tuberculosis can dephosphorylate phosphatidylinositol-4-monophosphate and phosphatidylinositol-(4,5)-bisphosphate to disrupt the localization of the N-terminus of GSDMD in the cell membrane, suggesting a novel mechanism for the regulation of pyroptosis [70].

The caspase-3-induced pyroptotic pathway has crosstalk with mitochondrial apoptosis. The leakage of cytochrome c triggered by cytotoxic chemotherapy is involved in the formation of the apoptosome, thereby cleaving pro-caspase-3 to cleave GSDME for the release of its N-terminus containing PFD. Subsequently, the N-terminus of GSDME undergoes oligomerization and re-localizes to the cell membrane, thus converting intrinsic apoptosis to pyroptosis or secondary necrosis [71]. Simultaneously, the caspase-8-induced pyroptotic pathway can be executed by the extrinsic apoptotic pathway. After the formation of complex II, caspase-8 is activated to cleave GSDMC, releasing its N-terminus containing PFD. The resulting GSDMC-N fragment translocates to the plasma membrane, converting extrinsic apoptosis to pyroptosis [72]. In addition, PD-L1 can also convert apoptosis to pyroptosis by cleaving GSDMC in cancer cells under hypoxic conditions [73]. During the infection of yersiniosis, the Yersinia effector protein YopJ is able to facilitate the activation of caspase-8 and inhibit the activation of TGF-β-activated kinase 1 (TAK1), leading to the cleavage of GSDMD and the release of its N-terminus containing PFD, causing pore formation in the cell membrane and pyroptosis [74,75]. Additionally, the cysteine protease SpeB of Streptococcus can directly drive the proteolytic cleavage of GSDMA to release the N-terminus containing PFD, resulting in pore formation in the plasma membrane and pyroptosis [76]. Moreover, granzyme A from cytotoxic lymphocytes and granzyme B from natural killer (NK) cells can cleave GSDMB and GSDME, respectively, contributing to the release of their N-terminus and the activation of pyroptosis [77,78]. Meanwhile, GSDMB can be divided into six isoforms due to alternative splicing, and only GSDMB isoform 3 and isoform 4 are capable of recognizing granzyme A to induce pyroptosis because isoform 3 and isoform 4 have a belt motif, raising questions regarding the discovery of GSDMB alternative splicing mechanisms among various diseases [79]. Figure 1 offers a comprehensive overview of the process of pyroptosis.

### 2.4. Ferroptosis

The term ferroptosis was first coined in 2012. As the name indicates, ferroptosis refers to a form of iron-dependent programmed cell death caused by iron overload in cells [80]. Specifically, iron is an essential trace element that maintains intracellular homeostasis, being involved in the transportation of oxygen, ATP generation, and DNA biosynthesis [81,82]. Under normal conditions, the membrane-bound protein transferrin receptor 1 (TFR1), which contains two ferric iron molecules, recognizes extracellular transferrin (TF) carrying ferric iron and imports iron into cells by triggering clathrin-dependent endocytosis of the entire holo-complex. The ferric iron is then transported into endosomes for reduction to ferrous iron by the STEAP (six-transmembrane epithelial antigen of the prostate) family of metalloreductases [83]. Subsequently, ferrous iron is released from the endosome into the cytosol via natural resistance-associated macrophage protein 2 (NRAMP2), and TFR1 is re-localized to the cell surface to uptake additional TF [84]. Ferritin, an iron-storage protein, can dynamically modulate the oxidation of intracellular ferrous iron to its ferric state, which is used in intracellular enzymatic reactions or stored for later use. Meanwhile, iron-saturated ferritin is degraded by nuclear receptor coactivator 4 (NCOA4)-mediated autophagy, termed ferritinophagy, to release its iron content [83,85]. Correspondingly, ferroportin is the only known iron exporter in mammalian cells, preventing the accumulation of excessive iron in the cell [86,87,88]. 

However, dysregulation of iron metabolism can cause an overload of ferrous iron in cells, inducing excessive generation of reactive oxygen species (ROS) through the Fenton reaction or oxidation of iron-binding enzymes. This results in the generation of fatal lipid peroxidation products, such as PL-hydroperoxide (PLOOH), malonaldehyde (MDA), and 4-hydroxynonenal (4-HNE) [88,89]. Lipid peroxidation is the final executor that induces cell damage and ferroptosis. In addition, the imbalance between the formation of oxidants and antioxidants triggers an abnormal redox system, which also contributes to the accumulation of lipid peroxides and ferroptosis. To be more specific, cells have evolved multiple antioxidant signaling cascades to protect themselves from ferroptosis, such as the cystine–glutamate antiporter (system Xc^−^) comprising two subunits, SLC3A2 (solute carrier family 3 member 2) and SLC7A11 (solute carrier family 7 member 11), the glutathione peroxidase 4 (GPX4) pathway, the ferroptosis suppressor protein 1-coenzyme Q10 (FSP1-CoQ10) pathway, the dihydroorotate dehydrogenase (DHODH)-CoQ10 pathway, and the GTP cyclohydrolase 1 (GCH1)-tetrahydrobiopterin (BH4) pathway [90,91,92,93]. The final products of the above pathways are glutathione (GSH), CoQ10H2, and BH4, which have ferroptotic protective effects because they are able to decrease the level of oxidative stress [94]. Conversely, dysregulation of the above pathways contributes to the initiation of ferroptosis. Recently, research identified that cyclic GMP-AMP synthase (cGAS) anchored to the outer mitochondrial membrane can associate with dynamin-related protein 1 (DRP1) to facilitate its oligomerization, leading to decreased levels of mitochondrial ROS and avoidance of ferroptosis [95]. Similarly, sex hormone receptors, such as the estrogen receptor (ER) and androgen receptor (AR), can upregulate the expression of phospholipid-modifying enzyme membrane-associated O-acetyl transferase genes (MBOAT1 and MBOAT2), which can reshape the cellular phospholipid profile to protect cells from ferroptosis [96]. Additionally, excessive activation of ferritinophagy and digestion of lipid droplets by autophagy, termed lipophagy, can contribute to lipid peroxidation and ferroptosis via intracellular iron overload and excessive production of free fatty acids, respectively [97,98,99]. In addition, novel clues indicate that selective autophagy, clockphagy, and chaperone-mediated autophagy can modulate the expression of modulators in the redox system to manipulate the process of ferroptosis [100,101]. Ferroptosis is regulated by various molecules. We summarize numerous crucial molecules involved in the modulation of cellular iron homeostasis and ferroptosis in Table 1 and describe the mechanisms of ferroptosis in Figure 2. In general, discovering novel effectors and mechanisms participating in the modulation of ferroptosis, or protective patterns of ferroptosis independent of the aforementioned mechanisms, can help us gain insight into the etiology of various diseases.

### 2.5. Cuproptosis

Cuproptosis was first identified in 2022 as a novel form of programmed cell death caused by abnormalities in systemic copper metabolism [154]. Specifically, copper ions are essential micronutrients present in all living mammals, especially in humans. They act as co-factors for enzymes, regulating the activity of various key metabolic enzymes and a broad range of physiological processes, such as mitochondrial oxidative phosphorylation (OXPHOS), tyrosine metabolism, neurotransmitter metabolism, redox reactions, extracellular matrix remodeling, and cell proliferation [7]. Meanwhile, the concentration of copper ions must be tightly regulated to ensure normal biochemical processes. Regarding systemic copper metabolism, the uptake of copper ions is mainly mediated by copper transporter 1 (CTR1, also called solute carrier family 31 member 1, SLC31A1) located on the apical side of the enterocytes in the small intestine. Subsequently, copper ions are translocated to the other side of enterocytes via copper chaperone antioxidant 1 (ATOX1) for release into the bloodstream through ATPase copper transporting alpha (ATP7A) and ATPase copper transporting beta (ATP7B) [155]. In the blood, copper ions prefer to bind with soluble chaperones, such as ceruloplasmin (CP), serum albumin (SA), transcuprein, histidines, and macroglobulins, rather than remaining free, and are subsequently transported to the liver by the portal system [7,155]. Upon absorption by the liver, copper ions bind with metallothionein 1/2 (MT1/2) and other thiol-rich proteins in a pH-dependent manner via their cysteine residues for storage or later use [156]. Simultaneously, ATPase copper transporting beta (ATP7B) in hepatocytes pumps copper ions back into the blood or bile, targeting specific tissues and organs or preventing the excessive accumulation of copper ions in the liver, respectively [157,158]. Upon reaching target tissues and organs, copper ions are internalized into the cytosol of target cells by SLC31A1 and then transported to the trans-Golgi network (TGN) or nucleus by chaperone antioxidant-1 (ATOX1) to facilitate the synthesis of cuproenzymes, including lysyl oxidase, tyrosinase, and ceruloplasmin, or to regulate gene expression related to cell proliferation [7]. In addition, copper ions can also be transported into mitochondria through the Cu chaperone for the superoxide dismutase (CCS)-superoxide dismutase 1 (SOD1) axis or cytochrome oxidase 17 (COX17) for detoxification of ROS and oxidative phosphorylation [7,157,159]. Additionally, intracellular ATP7A and ATP7B are responsible for exporting copper ions to prevent their accumulation in cells [7]. 

Nevertheless, abnormal systemic or intracellular copper metabolism, especially dysfunction of modulators, can cause the accumulation of extracellular divalent copper ions (Cu^2+^). Cu^2+^ can form a complex by binding with Elesclomol (ES) or Disulfiram (DSF) to internalize into the cytosol [154,160]. The cytosolic Cu^2+^ enters mitochondria and undergoes reduction from Cu^2+^ to the toxic monovalent copper ion (Cu^+^) via catalysis by ionophore ferredoxin 1 (FDX1) [154]. Significantly, Cu^2+^ is transported from the inner membrane of mitochondria into the mitochondrial matrix via solute carrier family 25 member 3 (SLC25A3), but the mechanism of how Cu^2+^ crosses from the outer membrane to the inner membrane of mitochondria remains unclear [161]. Cu^+^ can attach to fatty acylated proteins of the tricarboxylic acid (TCA) cycle, further inducing the aggregation of lipoylated proteins and depletion of iron-sulfur (Fe-S) proteins, resulting in protein toxicity, intracellular stress, mitochondrial shrinkage, membrane rupture, and, ultimately, cuproptosis [154,162]. Systemic and intracellular copper metabolisms, as well as cuproptosis, are described in Figure 2.

### 2.6. Other Forms of PCD

Mitochondrial permeability transition (MPT) was first described in 1976. MPT-driven necrosis is characterized as a unique form of programmed cell death (PCD), manifesting necrotic morphology, initiated by specific perturbations of the intracellular microenvironment such as severe oxidative stress and cytosolic calcium ion (Ca^2+^) overload [163]. MPT-driven necrosis relies on the activation of cyclophilin D (CYPD) and the formation of a supramolecular complex termed the permeability transition pore complex (PTPC) in the intermembrane space (IMS), leading to an abrupt loss of impermeability of the inner mitochondrial membrane (IMM) to small solutes and a rapid dissipation of mitochondrial potential [164,165]. The abnormality of mitochondria causes cell swelling, rupture of cell membranes, and eventual cell death [165]. However, the detailed mechanisms of MPT-driven necrosis need further investigation. Particularly, several factors, such as BAX, BAK, BID, dynamin 1-like (DRP1), and p53, can regulate MPT-driven necrosis by manipulating the activation of CYPD and PTPC formation, indicating crosstalk among intrinsic apoptosis, mitophagy, the cell cycle, and MPT-driven necrosis [1].

Autophagy is an evolutionarily conserved, intracellular, self-protective mechanism that maintains energy balance in response to nutrient stress in cells. The activation of autophagy can also degrade misfolded or aggregated proteins, damaged organelles, and invading pathogens, thereby maintaining intracellular homeostasis [166,167]. Generally, autophagy-dependent cell death (ADCD), first observed in 2006, is a distinct form of programmed cell death (PCD) that occurs due to the abnormal stimulation of autophagy in specific developmental or pathophysiological contexts and relies on the autophagic machinery or its components [168,169]. ADCD can be triggered by three distinct mechanisms: excessive ER-phagy, excessive mitophagy, and autosis [170]. The endoplasmic reticulum (ER) is the largest organelle in eukaryotic cells, responsible for the folding and trafficking of proteins that enter the secretory pathway by assembling a complex cell quality-control network [171]. Stress conditions, such as the unfolded protein response (UPR), lack of nutrients or oxygen, and pharmacologic stimuli, can induce ER-phagy by interacting with a series of ER-phagy receptors to remove damaged ER by delivering ER fragments to lysosomes [170,172]. However, the continuous degradation of ER fragments by ER-phagy can cause the excessive formation of autophagosomes, eventually leading to cell death [173,174]. Mitophagy is a conserved intracellular process that ensures mitochondrial quality and quantity control [175]. Mitophagy is initiated by specific mitochondrial outer membrane receptors interacting with proteins on the mitochondrial surface, leading to the formation of autophagosomes surrounding mitochondria to degrade damaged or depolarized mitochondria [176]. Nevertheless, excessive mitophagy results in an abnormal mitochondrial membrane potential and cell death [177]. Autosis was first observed by treating HeLa cells with the autophagy-specific activator BECN1-derived peptide (Tat-Beclin 1) [178]. Tat-Beclin 1 can directly cause the perturbation of sodium (Na^+^) and potassium (K^+^)-adenosine triphosphatase (ATPase) circulation, ultimately resulting in changes in autophagic flux and cell death [179]. Additionally, the process of autosis manifests in a time-dependent manner. During the early phase of autosis, the number of autophagosomes increases significantly while focal nuclear concavities occur. Subsequently, focal ballooning of the perinuclear space (PNS) and the disappearance of subcellular organelles appear in the later phase [180]. Additionally, autophagy is associated with numerous forms of PCD, such as apoptosis, necroptosis, ferroptosis, and cuproptosis [170,181].

Lysosomes are dynamic, single-membrane, and heterogeneous organelles that contain a wide variety of hydrolytic enzymes for the digestion of toxic intracellular components and damaged organelles, as well as for the termination of signal transduction [182]. Lysosome-dependent cell death (LDCD), first coined in 2000, is a form of programmed cell death (PCD) initiated by lysosomal membrane permeabilization (LMP), leading to the leakage of lysosomal contents into the cytosol, such as proteolytic enzymes of the cathepsin family and iron [5,183]. Subsequently, these leaked contents engage with apoptotic effectors, such as the p53 effector DNA damage-regulated autophagy modulator 1 (DRAM1), leading to mitochondrial outer membrane permeabilization (MOMP) and the activation of caspase-dependent signaling, eventually causing cell death [5].

Parthanatos, a poly(ADP-ribose) polymerase 1 (PARP1)-dependent form of programmed cell death (PCD), was first described in 2008 [184]. The accumulation of cytotoxic stimuli, such as oxidative stress, hypoxia, hypoglycemia, and inflammatory conditions, can cause PARP1 hyperactivation, resulting in the depletion of nicotinamide adenine dinucleotide (NAD^+^) and adenosine triphosphate (ATP), as well as the aggregation of polymers and poly(ADP-ribosyl)ated proteins at mitochondria, ultimately causing cell death due to mitochondrial membrane dissipation and mitochondrial outer membrane permeabilization (MOMP) [185].

Alkaliptosis, first unveiled in 2018, is caused by intracellular alkalinization, which involves the suppression of the NF-κB (nuclear factor κB)-carbonic anhydrase 9 (CA9) pathway and the ATP6V0D1-STAT3 pathway [186,187]. This dysregulation of intracellular pH results in cell death.

Oxeiptosis, first termed in 2018, is an oxygen radical-induced form of programmed cell death (PCD) initiated by the hyperactivation of the KEAP1-PGAM5-AIFM1 signaling cascade [188].

Disulfidptosis is a novel form of programmed cell death (PCD), defined in 2023, initiated by glucose starvation in cells with high SLC7A11 expression [189]. High uptake of cystine, coupled with a shortage of nicotinamide adenine dinucleotide phosphate (NADPH) supply, results in NADPH depletion, aberrant disulfide binding to actin cytoskeleton proteins, actin network collapse, and subsequent cell death [189].

NETs are extracellular net-like DNA–protein structures released by cells in response to various cellular stresses, including pathogen infections or injuries. They can also be formed by other leukocyte types, such as mast cells, eosinophils, and basophils, as well as epithelial cells and cancer cells [5]. The term NETosis was first coined in 2004 and was observed in neutrophils upon exposure to phorbol myristate acetate or interleukin 8 (IL-8), describing the process of NET generation [190]. At the molecular level, NETosis is a dynamic process involving multiple signaling pathways, such as NADPH oxidase-mediated ROS production, protein kinase C (PKC) isoform-initiated signaling cascades, autophagy, the release and translocation of granular enzymes, and the trafficking of N-GSDMD from the cytosol to the nucleus. This leads to various abnormal biological processes, including histone citrullination, chromatin decondensation, the destruction of the nuclear envelope, the release of chromatin fibers, and the formation of pores in the plasma membrane [191,192,193,194,195]. 

ENTosis, first introduced in 2007, is characterized by one cell inserting itself into a neighboring cell, a process termed the cell-in-cell (CIC) pattern, ultimately causing the death of the invading cell [196]. Glucose starvation, matrix deadhesion, and mitotic stress can induce ENTosis through cell adhesion and cytoskeletal rearrangement pathways [196,197,198,199,200]. Although the underlying mechanisms of ENTosis are not well understood, adhesion proteins such as cadherin 1 (E-cadherin), catenin alpha 1 (CTNNA1), and microtubules play a central role in the formation of adherent junctions between cells, leading to the generation of CIC structures and cell death [5]. The aforementioned forms of PCD are described in Figure 3 and Figure 4.

### 2.7. Characteristics of Different PCD Subroutines

Various PCD subroutines cause the disintegration of cells through distinct signaling cascades, resulting in differing morphological changes and immunological consequences. PCD can be classified into immunogenic cell death (ICD), also known as lytic forms of cell death, and tolerogenic cell death (TCD), also known as non-lytic forms of cell death [201,202]. ICD induces the activation of the immune system, whereas TCD does not provoke any inflammatory or immune reactions. ICD elicits acute or chronic inflammatory responses by releasing DAMPs from dead or dying cells into extracellular spaces. These DAMPs are subsequently recognized by pattern recognition receptors (PRRs) or other receptor systems expressed by neighboring macrophages and other bystander cells, triggering the release of proinflammatory cytokines [203]. Additionally, DAMPs play a fundamental role in regulating the balance between ICD and TCD [5]. ICD not only promotes tissue regeneration and organ development but also contributes to the progression of inflammation in numerous human diseases, especially neurodegenerative disorders. Therefore, agents that inhibit PCD may be critical components of future clinical therapeutic strategies. We summarize the immune and morphological hallmark features and major inhibitors of various PCD modalities in Table 2.

## 3. PCD in NDDs

### 3.1. PCD in AD

Alzheimer’s disease (AD), first described by Alois Alzheimer in 1906, is the primary cause of dementia. It has now become one of the most expensive, lethal, and burdensome diseases, with huge implications for individuals and society [207]. The onset of AD is relatively insidious, characterized by substantial progressive cognitive impairment and memory loss associated with age, impacting daily life functionality [208]. Amyloid precursor protein (APP) is widely present in the endoplasmic reticulum (ER) of neurons and glial cells, mediating neurotransmitter release, cell-to-cell adhesion, and neuronal signaling [209]. The cleavage of APP by α-secretase and γ-secretase produces non-toxic, soluble, and neuroprotective APPα peptides, whereas APP cleaved by β-secretase and γ-secretase produces neurotoxic amyloid β (Aβ) oligomers [210]. The aggregation of Aβ oligomers between nerve cells causes the formation of Aβ plaques, leading to neuronal cell death, particularly in the hippocampus [17]. Meanwhile, type 2 microtubule-associated protein (Tau) is expressed in neurons, astrocytes, and oligodendrocytes and is responsible for stabilizing microtubule structures by directly binding to them [211]. However, hyperphosphorylation of Tau due to abnormal post-translational modifications leads to the dissociation of microtubules and the aggregation of neurotoxic Tau proteins [212]. Aβ deposition and Tau aggregation facilitate the generation of neurofibrillary tangles (NFTs) in the cortex, leading to the progression of AD [213]. Generally, the formation of Aβ plaques, Tau aggregation, and NFT formation in neuronal cells are primary features of AD.

The effectors and signaling cascades of programmed cell death (PCD) play an essential role in the onset and progression of Alzheimer’s disease (AD). Specifically, the formation of the neurotoxic Aβ peptide can also be mediated by caspase-3 instead of β-secretase and γ-secretase, and members of the caspase superfamily can be activated by Aβ [214]. During NFT formation, anti-apoptotic factor expression is restrained, while proapoptotic protein levels are elevated via p53-dependent transcriptional upregulation [214,215]. Extracellular Aβ deposition can be recognized by apoptotic death receptors, leading to the activation of extrinsic apoptotic pathways [216]. Intracellular Aβ can insert into the mitochondrial outer membrane, leading to the formation of the mitochondrial permeability transition (MPT) and subsequent leakage of cytochrome c, causing mitochondrial apoptosis in neurons [192]. Aβ in the endoplasmic reticulum (ER) can also cause ER stress and initiate caspase-12/caspase-7-induced apoptosis in neurons [217]. Additionally, MAPK, JNK, BDNF-TrkB-CREB, JAK-STAT, PI3K-Akt-mTOR, and GSK-3β pathways are involved in the formation and aggregation of Aβ, as well as the hyperphosphorylation and aggregation of Tau through interactions with apoptotic signaling pathways [214]. Concurrently, the necroptotic RIPK1-RIPK3 complex facilitates the formation of Aβ structures, aiding in the translocation of Aβ to the cell surface and its aggregation [218]. Aβ plaques can stimulate microglia to secrete inflammatory factors, including TNF-α, thereby inducing the pyroptosis of neurons [219]. Hyperphosphorylated Tau can simultaneously activate necroptosis and the NF-κB pathway, contributing to the formation of NFTs and cytokine storm in microglia [220]. Similarly, Aβ and hyperphosphorylated Tau mediate the activation of the NLPR3-caspase 1-GSDMD axis and the release of caspase-1, IL-1β, and IL-8, leading to the activation of pyroptosis and formation of NFTs, ultimately causing chronic inflammation in microglia [221,222,223]. Notably, the effectors of ferroptotic antioxidant signaling cascades exhibit decreased expression in the neurons of AD patients, suggesting that ferroptosis may be a significant mechanism in AD [224]. Iron overload in the brain can exacerbate the production and accumulation of Aβ by enhancing the activity of β-secretase [225]. Moreover, Aβ can bind to ferrous iron and subsequently initiate lipid peroxidation and ferroptosis via the Fenton reaction [225]. Additionally, ferric iron can directly bind to Tau, causing its hyperphosphorylation and accumulation, along with increased expression of HO1 [226,227]. Comparatively, abnormal accumulation or deficiency of copper ions due to dysregulation of copper metabolism can also be observed in specific areas of the brain in AD patients [224]. A molar ratio of Cu^2+^ to Aβ oligomers of 0.25:1 can facilitate Aβ plaque formation [228] and can also be associated with the Tau R1 peptide to regulate Tau aggregation, while the Tau R2 peptide can reduce Cu^2+^ to Cu+ to induce cuproptosis [7]. Cu^2+^ can also trigger the NF-κB signaling pathway, increasing the release of inflammatory factors in microglia and impairing the brain’s ability to remove Aβ peptides by reducing the expression of lipoprotein receptor-related protein 1 (LRP1) [229,230]. Aβ oligomers in the mitochondria can activate CYPD by directly binding, causing mitochondrial perturbation and potentially activating MPT-driven necrosis in the neurons of the temporal cortex and hippocampus [231,232]. The accumulation of autophagosomes and lysosomes, along with higher expression of autophagy-related genes associated with increased levels of Aβ, has been observed in hippocampal CA1 pyramidal neurons and other neurons, eventually leading to cell death [233,234]. Aβ peptides in the hippocampus and microglia can also induce nitric oxide (NO) production, triggering DNA damage and PARP1 activation, potentially resulting in Parthanatos, neuroinflammation, and alterations in hippocampal synaptic integrity [235,236]. During the progression of AD, circulating neutrophils in the peripheral blood can be recruited to the vessel walls of the CNS via the LFA-1 β2 integrin–ICAM-1 complex released from cerebral endothelial cells. Subsequently, β2 integrins are activated, causing neutrophil arrest and the formation of NETs, resulting in damage to the blood–brain barrier, which is positively associated with Aβ depositions [237]. Figure 5 illustrates various PCD modalities in AD. 

### 3.2. PCD in PD

Parkinson’s disease (PD), first described by James Parkinson in 1817, is a complex, progressive, and multisystem neurodegenerative disease with a range of causes and clinical presentations, elicited by the combined effects of environmental and genetic factors [238]. Clinical syndromes of PD include pathological motor features, such as a slowly progressive asymmetric resting tremor, cogwheel rigidity, and bradykinesia, as well as non-motor features, including anosmia, constipation, depression, sleep behavior disorder, autonomic dysfunction, pain, cognitive decline, and psychiatric symptoms [239]. α-synuclein (αSyn), consisting of amphipathic alpha-helical repeats, is an abundant neuronal protein enriched at synapses and mediating neurotransmission [240]. Pathologically, αSyn can undergo conformational changes capable of aggregation due to mutations in its encoding gene Alpha-synuclein (SNCA). Aggregated αSyn proteins then act as a major component of Lewy bodies and Lewy neurites, which are typical hallmarks of PD [241]. Significantly, Lewy bodies contain hundreds of other proteins and dysmorphic organelles, including lysosomes and mitochondria, packaged by abundant lipid membranes [242]. Additionally, the loss of dopamine neurotransmission due to the death of dopaminergic neurons projecting from the substantia nigra pars compacta to the caudate-putamen in the striatum is another leading cause of PD [241]. Furthermore, mutations in genes responsible for maintaining mitochondrial quality in neuronal cells, such as parkin RBR E3 ubiquitin protein ligase (PRKN), leucine-rich repeat kinase 2 (LRRK2), PTEN-induced kinase 1 (PINK1), and Parkinsonism-associated deglycase (PARK7), are associated with inherited PD [243].

In Parkinson’s disease (PD), intrinsic apoptosis is the predominant driver of dopaminergic neuronal death. Numerous pieces of evidence suggest that mutated PRKN, LRRK2, PINK1, and PARK7 localize in the outer mitochondrial membrane of Lewy body-positive neurons and can cause the disruption of mitochondrial outer membrane permeability (MOMP) and the formation of mitochondrial permeability transition (MPT), ensuing leakage of cytochrome c and activation of intrinsic apoptosis [243,244]. Additionally, observations of brain tissue from PD patients illustrate the abnormally increased expression of caspase 3 and BAX, along with reductions in Bcl-2 superfamily protein levels, which are associated with the upregulation of p53 [245,246,247]. The aggregation of αSyn in dopaminergic neurons can disrupt mitochondrial homeostasis, making dopaminergic neurons prone to apoptosis [248]. Meanwhile, microglial neuroinflammation in PD is induced by the formation of αSyn-derived NLRP3 inflammasomes and pyroptosis in dopaminergic neurons [249]. Mutated LRRK2 can also induce the activation of gasdermin D (GSDMD) to facilitate the release of reactive oxygen species (ROS) and necroptosis [250]. In the postmortem substantia nigra of individuals with PD, the expression of receptor-interacting protein kinase 1 (RIPK1), RIPK3, and mixed-lineage kinase domain-like protein (MLKL) is elevated, and mutated LRRK2 proteins are positively associated with the activation of MLKL and necroptosis [251,252]. Notably, the induction of ferroptosis is highly synchronized with the progression of PD. Ferric iron and αSyn coexist in the Lewy bodies of PD patients, and ferric iron is essential for the aggregation of αSyn [253,254]. Iron accumulation in the brain can activate microglia to release proinflammatory cytokines and cause oxidative stress, leading to the ferroptosis of dopaminergic neurons [255]. Comparatively, the N-terminal of αSyn contains a copper-binding site, and abnormally increased or decreased copper concentrations are associated with the progression of PD [7,256]. Moreover, an independent study illustrated that the ablation of cyclophilin D (CYPD) in PD-linked αSyn mutant transgenic mice delayed disease onset and extended lifespan, suggesting that CYPD may induce MPT-driven necrosis to regulate PD development [257]. The activation of autophagy and dysregulation of lysosomes in dopaminergic neurons can also be observed in PD patients [258,259]. The activation of poly (ADP-ribose) polymerase 1 (PARP1) can increase the neurotoxicity of αSyn by changing its conformation, inducing parthanatos [260]. The different PCD modalities in PD are comprehensively described in Figure 5.

### 3.3. PCD in HD

Huntington’s disease (HD), first described by George Huntington in 1872, is an autosomal dominantly inherited neurodegenerative disorder characterized by progressive motor, behavioral, and cognitive decline with high mortality [261]. The huntingtin (HTT) protein is present in spindle poles and microtubules, regulating cell division, ciliogenesis, endocytosis, transcription, vesicular transport, and autophagy [262]. The pathology of HD is monogenic and characterized by the production of mutant HTT proteins with an abnormally long polyglutamine repeat due to CAG trinucleotide repeat expansion in the HTT gene on chromosome 4, which eventually results in the aggregation of mutant huntingtin (mHTT) proteins in neurons and glial cells, particularly GABAergic and motor neurons [262]. As HD progresses, significant neuronal death can be observed in cortical, thalamic, and hypothalamic areas and even the entire brain, along with atrophy of the basal ganglia [263].

mHTT proteins can significantly facilitate the expression of pro-apoptotic factors, such as BIM and BAX, and apoptosis, while loss of BIM can decrease the amounts of mHTT proteins and neuronal cell death, indicating that pro-apoptotic factors potentially participate in the generation of mHTT proteins [264,265,266]. Additionally, cleaved caspase-3 can cleave mHTT proteins to produce more neurotoxic fragments that translocate into the nucleus and subsequently interact with different transcription factors, such as p53, eventually causing mitochondrial disruption, while wild-type (WT) HTT protein can prevent caspase-3 activation [267,268,269]. Meanwhile, mHTT proteins can cause mitochondrial dysfunction and the subsequent release of cytochrome c [270,271]. Moreover, a study confirms that mHTT proteins have the potential to activate receptor-interacting protein kinase 1 (RIPK1) and necroptosis in specific neurons, leading to neuroinflammation and the progression of HD [272]. Interestingly, the levels of caspase-1 and NLRP3 are intensely elevated in striatal spiny projection neurons and in parvalbumin interneurons, deteriorating the symptoms of HD [273]. Increased levels of toxic iron in the brain contribute to the HD process due to increased levels of reactive oxygen species (ROS) and depletion of glutathione peroxidase 4 (GPX4) in spinal motor neurons, showing the significant role of ferroptosis in HD progression [274,275]. Similarly, the abnormal elevation of copper concentrations in the brain contributes to the onset and progression of HD, and the wild-type (WT) HTT protein has two potential copper-binding residues [276]. Additionally, a similar scenario can be seen in HD, where the accumulation of mHTT is associated with attenuated autophagy [277]. Various PCD modalities in AD, PD, and HD are described in Figure 5. As shown in Figure 5, various PCD modalities are described in the context of HD.

### 3.4. PCD in ALS, SMA and MS

Amyotrophic lateral sclerosis (ALS), first described by Jean Martin Charcot in 1869, is a fatal neuromuscular disease characterized by progressive muscle weakness and atrophy due to the loss of both upper motor neurons (UMNs) and lower motor neurons (LMNs). This leads to patients experiencing dysphagia, dysarthria, and limb weakness, eventually dying from respiratory complications [278]. ALS can be classified into two types: familial ALS, which constitutes 10 to 15% of cases and is inherited, and sporadic ALS, which constitutes the remaining (approximately 85%) of cases [12]. Pathologically, ALS-associated genetic signatures vary in frequency, but the most common mutations occur in four genes: chromosome 9 open reading frame 72 (C9ORF72), TAR DNA-binding protein (TARDBP), superoxide dismutase 1 (SOD1), and fused in sarcoma/translocated in liposarcoma (FUS). These mutations can impair various intracellular functions and form protein aggregations, which accelerate UMN and LMN loss and contribute to the onset and progression of most ALS cases [279]. Spinal muscular atrophy (SMA), first identified by William R. Kennedy in 1966, is an autosomal recessive, progressive, and lethal neuromuscular disorder characterized by the degeneration of alpha motor neurons in the spinal cord [280].

Spinal muscular atrophy (SMA) can be clinically classified into four grades of severity (SMA I, SMA II, SMA III, and SMA IV) according to age of onset and motor function achieved [281]. Clinical hallmarks of SMA include muscular dystrophy, fasciculations, altered reflexes, joint contractures, dysphagia, dysarthria, and respiratory complications [282]. Survival motor neuron (SMN) protein is involved in the biogenesis of small nuclear RNA (snRNA) and ribonucleoproteins (snRNPs), which act as major components of the pre-mRNA splicing machinery [283]. More than 95% of SMA cases exhibit a homozygous deletion or point mutation in exon 7 of the SMN1 gene, leading to the loss of SMN production [284]. Additionally, SMN2, a paralogous copy of SMN1, has a single-nucleotide difference in exon 7 compared with SMN1, resulting in the alternative splicing of exon 7 in transcripts and the expression of only 5–10% of full-length functional SMN protein [285]. The copy number of SMN2 is inversely proportional to the age of onset and severity of SMA [286].

Multiple sclerosis (MS) is an immunological disease and a neurodegenerative condition that causes chronic inflammation and acute inflammatory lesions in the central nervous system (CNS), eventually resulting in tissue damage and disability [287]. MS often occurs in young populations, and the clinical manifestations of MS patients are highly variable, including optic neuritis, weakness or changes in sensation in the body, dizziness, memory or cognitive impairment, dysregulation of bladder control, and depression or anxiety [288]. MS can be divided into three representative types according to the onset of recurring clinical symptoms followed by total or partial recovery: relapsing–remitting MS (RRMS), primary progressive MS (PPMS), and secondary progressive MS (SPMS). These types are influenced by factors, such as low uptake of vitamin D, low levels of sunlight exposure, Epstein-Barr virus (EBV) infection, and genetic predisposition [288]. The two pathologic hallmarks of MS are axonal degeneration and neuronal cell death, which are induced by oxidative stress and mitochondrial dysfunction in active MS lesions [289,290].

Research has discovered that the expression of pro-apoptotic factors and the caspase superfamily is significantly elevated in the motor neurons of ALS and SMA patients, along with abnormally low levels of anti-apoptotic factors [291,292,293]. Meanwhile, apoptosis is widely induced in astrocytes, microglia, oligodendrocytes, and neurons in MS lesions, contributing to the progression of MS [294]. Specifically, mutant SOD1 can bind to the Bcl-2 anti-apoptotic factor, suppressing its activity [295]. Mutations in a highly conserved region of TARDBP can cause the formation of mutant forms of TAR DNA-binding protein 43 (TDP-43), which can induce neural apoptosis [296]. Loss of SMN contributes to the activation of p53 and JNK signaling pathways, subsequently inducing apoptosis [297,298]. Additionally, the expression of RIPK1 and RIPK3, as well as the formation of necrosomes, is enhanced in pathological tissues of SOD1 (G93A) ALS transgenic mice [299,300]. A similar scenario is observed in the neuronal cells of cortical lesions in the human MS brain [301]. Furthermore, an independent study identified that the knockout (KO) of RIPK3 can significantly increase the survival and motor function of SMN deletion mice [302].

The elevated formation of inflammasomes, expression of IL-1β and IL-18, and excessive cleavage of GSDMD have been reported in amyotrophic lateral sclerosis (ALS) and multiple sclerosis (MS) cases, indicating a crucial role for pyroptosis in the neuroinflammation and development of ALS [67,303,304]. Studies have also suggested that dysregulation of iron and copper homeostasis, as well as excessive reactive oxygen species (ROS) production, can be observed in the brains of ALS and MS patients, which may induce ferroptosis and cuproptosis, leading to neuronal damage [17,305,306,307,308]. Furthermore, several studies have revealed that cyclophilin D (CYPD) and mitochondrial permeability transition (MPT)-driven necrosis may contribute to the pathogenesis of ALS and MS. For example, mutant SOD1 can interact with CYPD to cause CYPD hyperactivation, thereby inducing the formation of mitochondrial permeability transition pore (mPTP) [309]. The activation of CYPD and formation of mPTP can also be observed in the axonal damage occurring during MS, which weakens the resistance to reactive oxygen and nitrogen species, thereby mediating axonal damage [310]. In addition, FUS can be recruited by PARP-1, activated by DNA damage, to stimulate the synthesis of long poly (ADP-ribose) (PAR) chains, indicating a role for parthanatos in ALS [311]. C9ORF72 deficiency or mutations in the brains of ALS patients may exacerbate the accumulation of DNA damage and PARP1 overactivation, leading to the activation of parthanatos [312]. Excessive PARP1 hyperactivation and parthanatos can also be detected in oligodendrocytes, astrocytes, and microglia or macrophages in the active areas of brain lesions in MS patients [313]. The abnormal accumulation of autophagosomes in the neuronal cells of ALS, SMA, and MS patients has been reported. The hallmark of autophagy-dependent cell death (ADCD) is the excessive formation of autophagosomes, revealing that ADCD is another form of neuronal death in the context of ALS [314,315,316]. The activation of neutrophils and the subsequent formation of neutrophil extracellular traps (NETosis) are elevated during the occurrence and progression of ALS and MS [317,318,319]. We describe various PCD modalities in ALS, SMA, and MS in Figure 6.

### 3.5. PCD in TBI and Stroke

Traumatic brain injury (TBI) refers to a physical injury caused by an external mechanical force, which induces transitory or permanent damage to brain tissues [320]. The pathophysiology of TBI involves both primary and secondary injury mechanisms. Primary injury occurs at the moment of impact, causing immediate damage to brain tissue or brain structures. Secondary injury mechanisms involve a neurodegenerative process manifesting from hours to days following the initial trauma, inducing chronic inflammation and neuron loss [320]. The secondary mechanism exacerbates the initial damage caused by the primary injury, and clinical presentations of secondary TBI include a combination of cognitive, emotional, and behavioral changes [321]. Stroke is a chronic neurodegenerative disease caused by insufficient blood supply to the brain. It can be divided into two major types: ischemic stroke, caused by occlusion of carotid and vertebral arteries, and hemorrhagic stroke, caused by subarachnoid or intraparenchymal hemorrhage [322,323]. Notably, ischemic strokes account for the majority of all stroke cases [324]. Common clinical symptoms of stroke include sudden weakness or numbness on one side of the body, aphasia, dysphasia, dizziness, and severe headache [325]. The primary pathophysiology of stroke involves the damage and death of brain cells due to the interruption of blood flow to the brain, which restrains the oxygen and nutrients reaching brain cells [326].

Generally, programmed cell death (PCD) plays an important role in the later stages of traumatic brain injury (TBI) and stroke progression, impacting the recovery of brain tissue and neurological function. Specifically, the formation of apoptotic bodies can be observed in post-ischemic stroke neurons, and apoptosis in neurons of the ischemic penumbra may be recoverable [327,328]. Additionally, neurons alter the glucose metabolism pathway from aerobic oxidation to anaerobic oxidation to deal with glucose starvation during stroke, leading to a lower production of ATP [329]. This change can impair Na^+^/Ca^2+^ influx and K^+^ efflux, causing the accumulation of intracellular Ca^2+^ and calpain activation, which leads to the cleavage of the anti-apoptotic protein Bcl-2 and intrinsic apoptosis [330]. Importantly, glucose starvation initiates disulfidoptosis, and research has discovered that disulfidoptosis-related genes (DRGs) are significantly associated with stroke in immune cells from peripheral blood samples of stroke patients [331,332]. Moreover, DNA damage-associated activation of the p53 signaling pathway and large amounts of ROS generation also contribute to the activation of intrinsic apoptosis during stroke progression [333]. Additionally, in the early stages of ischemic stroke, the activation of immune cells, such as microglia, can release TNFα and FasL, engaging death receptors to initiate extrinsic apoptosis [334,335]. Similarly, Ca^2+^ overload, DNA damage, excessive ROS generation, and activation of immune cells can also be observed in the CNS of TBI patients, activating intrinsic and extrinsic apoptosis of neuronal cells [336]. However, necroptosis may occur even if the apoptotic signal is suppressed during stroke and TBI progression. Research has identified that the secretion of TNF-α, TRAIL, and FasL by microglial cells can be recognized by death receptors on neurons, triggering RIPK1-RIPK3-MLKL signaling cascade activation and necrosome formation [45]. Meanwhile, microglial cells also secrete proinflammatory cytokines, such as IL-1β, causing caspase-1-mediated pyroptosis in neuronal cells under TBI and stroke conditions [337,338]. Additionally, iron accumulation and lipid peroxidation can be observed in multiple areas of the brains of TBI and stroke patients due to the loss of antioxidant signaling cascades [333,339]. Similarly, the disruption of copper homeostasis caused by insufficient ATP generation can also be observed in the CNS of TBI and stroke patients, and cuproptosis-related genes regulate immune infiltration in ischemic stroke [340,341,342]. Moreover, hyperactivation of PARP1 and depletion of NAD^+^ and ATP can be observed in stroke and TBI mouse models, triggering parthanatos in neurons [343,344]. Meanwhile, ATP depletion in neurons of TBI and stroke mouse models can also induce the activation of CYPD-dependent MPT-driven necrosis, while hypoxia-induced activation of p53 can interact with CYPD to exert an anti-angiogenic effect in the brain after ischemic stroke [345,346,347]. Additionally, excessive activation of the hypoxia-inducible factor 1α (HIF-1α) signaling pathway and ER stress mTOR signaling pathways can be observed in microglia and neurons in the brains of stroke patients, subsequently activating ADCD [323]. Similarly, excessive mitophagy and ER-phagy, as well as autosis, can also be observed in immune cells and neurons in the brains of TBI patients [348]. LDCD of endothelial cells can also be detected in stroke patients, potentially causing damage to the blood–brain barrier (BBB) [15]. Additionally, the release of high-mobility group box 1 (HMGB1) by platelets can facilitate NETosis in the acute phase of stroke, exacerbating disease progression [349]. During stroke development, NETosis generated by neutrophils exacerbates neuroinflammation and impairs revascularization and vascular remodeling after stroke due to the upregulation of peptidylarginine deiminase 4 (PAD4) [350]. NETosis also contributes to coagulopathy and neuroinflammation after TBI through the release of HMGB1 by platelets and the formation of neutrophil–platelet aggregates [351,352]. Various PCD modalities in stroke and TBI are illustrated in Figure 7.

### 3.6. Therapeutic Strategies Targeting PCD Signaling Pathways in NDDs

Following this review, it is evident that multiple programmed cell death (PCD) pathways collectively play a role in neurodegenerative diseases (NDDs). The regulation of PCD occupies a significant position in the complex pathogenesis of NDDs. As research continues to uncover the mechanisms underlying disease progression, our understanding of targeting PCD to modulate NDDs has deepened. Specifically, numerous key factors involved in PCD pathways have been identified as the most direct and promising targets for therapeutic development. These include genes and proteins, such as the caspase family, apoptosis-related factors, necrosome-related factors, and inflammasome-related factors, each of which directly influences PCD processes in NDDs. Furthermore, the development of PCD pathways in NDDs is often accompanied by various associated biological phenomena. Strategies targeting processes such as oxidative stress, neuroinflammation, and metabolic imbalance also hold great promise and therapeutic value for disease treatment. This review discusses the targets and strategies for treating NDDs by focusing on PCD pathways.

As major participants in cell death pathways, the caspase family is extensively involved in both apoptotic and non-apoptotic cell death processes. Inhibiting the caspase family to interrupt PCD processes mediated by them has been demonstrated in numerous studies to be beneficial for improving and treating symptoms of NDDs. Caspases with short prodomains (Caspase-3, -6, and -7) are known as effectors of apoptosis, while those with long prodomains can be further classified into initiators of apoptosis (Caspase-2, -8, -9, and -10) and inflammatory caspases that cleave cytokines (Caspase-1, -4, -5, -11, and -12) [353]. Caspase-2 may function as both an initiator and an effector of apoptosis [353]. The expression level of caspase-3 is significantly higher in Alzheimer’s disease (AD) patients compared to age-matched controls. Inhibiting caspase-3 activity can alleviate Alzheimer-like phenotypes in transgenic mice, and pharmacological experiments in AD models have demonstrated clinical improvements that mitigate symptoms [354,355,356,357]. Furthermore, inhibiting the JAK-STAT-caspase-3 axis to prevent neurodegenerative diseases has also been reported [214,358]. Caspase-6 is another factor considered a potential diagnostic marker and therapeutic target for Huntington’s disease (HD) patients. Enhancing the insulin-like growth factor 1 (IGF-1) signaling pathway reduces HTT toxicity changes associated with increased caspase-6 activation, and IGF-1 treatment has demonstrated therapeutic benefits in HD mouse models [356,359,360,361]. Additionally, the co-expression of caspase-2 and Bcl-2-interacting mediator of cell death (Bim) has been observed in neurons of AD brains [362]. Studies have shown that caspase-2 deficiency can ameliorate spine density reduction in the J20 mouse model of AD, preventing behavioral changes in these mice [363]. In Alzheimer’s disease and related dementias (ADRD), Tau is cleaved by caspase-2, producing Δtau314, which promotes Tau mislocalization and accumulation (Casp2/tau/Δtau314). Inhibiting caspase-2 as a drug target can ameliorate synaptic dysfunction in ADRD [364]. The inhibitor of the apoptosis protein (IAP) family, as endogenous inhibitors of the caspase family, suppresses cell death by directly acting on caspases or serving as targets for protein degradation [353,365]. In spinal muscular atrophy (SMA), neuron-specific IAP family members NAIP and XIAP effectively block the enzymatic activity of group II caspases (3 and 7) and reduce the expression levels of cleaved-caspase-3, thereby protecting spinal motor neurons (MNs) and preventing severe SMA [366,367]. In a mouse stroke model, early ischemic activation of the apoptotic pathway in the striatum is associated with caspase-9 activation. Treatment with the endogenous caspase-9 inhibitor XIAP-BIR3 has been shown to protect neuroanatomy and function in the disease model [353]. As critical anti-apoptotic proteins, Bcl-2 family proteins exhibit suppressed expression across various NDDs, profoundly influencing the progression of these conditions. The practical role and physiological significance of targeting Bcl-2 family proteins to regulate the onset and development of NDDs have been extensively documented in numerous studies [368]. 

Strategies employing Bcl-2 family-mediated apoptosis to treat amyotrophic lateral sclerosis (ALS) have been validated in ALS mouse models; the knockout of two Bcl-2 proteins, Bax and Bak, counteracts the toxic effects of mutant superoxide dismutase 1 (SOD1) by inhibiting the activation of pro-caspase-3, thereby preventing neuronal damage [369,370]. Additionally, in HD, huntingtin (HTT) enhances the activity of caspase-8 and calpain, leading to the cleavage of full-length Bid. In the superior cervical ganglion (SCG), mutant HTT induces Bax-independent cell death [264,371,372].

Inflammation and oxidative stress are pervasive in NDDs and PCD, serving both as triggers and concomitant biological phenomena. These processes play a central role in the pathogenesis of NDDs as critical pathophysiological features [373,374,375,376]. The caspase family, which initiates apoptosis, is not only involved in regulating cell death but also plays a crucial role in modulating neuroinflammation in PCD. Numerous studies have demonstrated that inhibiting caspase-1 alleviates PD symptoms by suppressing neuroinflammation [377,378,379,380]. Research indicates that the NLRP3/caspase-1 axis and the gasdermin (GSDM) family represent substantial interactions between neuroinflammation and the initiation of apoptosis [381]. Inhibiting the NLRP3/caspase-1 axis reduces amyloid-beta deposition in the AD APP/PS1 model, and disease models in Nlrp3^−/−^ or Casp1^−/−^ mice also exhibit symptom relief [381]. Additionally, the therapeutic potential of inhibiting p38 through the suppression of the NLRP3 inflammasome pathway has been shown. p38 inhibitors such as SB203580 and NLRP3 inhibitors like MCC950 not only prevent neurodegeneration in vivo but also alleviate motor deficits in the α-Syn-A53T transgenic mouse model of PD [379]. Beyond these two inhibitors, Celastrol and the small-molecule kaempferol (Ka) also mitigate PD symptoms via the same pathway, showing similar trends in multiple sclerosis (MS) studies [377,378,382]. The overactivation of the NLRP3 inflammasome impairs microglial autophagy, exacerbating neurodegenerative disease mechanisms, thus supporting the application of microglial autophagy inducers and NLRP3 inhibitors [383]. In amyotrophic lateral sclerosis, knockdown of GSDME reduces neuroinflammation and rescues the loss of motor neurons derived from patient-induced pluripotent stem cells (iPSCs). Deleting GSDME in SOD1G93A ALS mice also shows effective therapeutic outcomes [377,384]. 

Furthermore, the regulation of oxidative stress in cell death pathways has been characterized in NDDs. The post-transcriptional regulator hsa-miR-4639-5p of PARK7 reduces PARK7 protein levels when upregulated, exacerbating oxidative stress and leading to neuronal death [385]. Silencing transcription factor RE1-silencing transcription factor (REST) and REST-dependent epigenetic remodeling have been reported to inhibit genes associated with oxidative stress and β-amyloid toxicity, thereby preventing neuronal death, which also plays a crucial role in other NDDs [386,387]. During acute neuronal insult events, hydrogen sulfide (H_2_S) acts as an antioxidant, anti-inflammatory, and anti-apoptotic mediator, protecting neurons from secondary neuronal damage [388]. In traumatic brain-penetrating injury (PTBI), a significant increase in reactive oxygen species (ROS) and reactive nitrogen species (RNS) production and elevated oxidative stress markers are observed, ultimately leading to cell death. Adjusting abnormal oxidative stress levels offers important insights for disease treatment and target development [389].

Microglial cells, which serve as critical sites for ion metabolism and neuroinflammation, contain various ion channels, including potassium (K^+^), calcium (Ca^2+^), chloride (Cl^−^), sodium (Na^+^), and proton (H^+^) channels. These ion channels are responsible for the dynamic characteristics of brain immune cells and play essential roles in regulating microglial proliferation, chemotaxis, phagocytosis, antigen recognition and presentation, apoptosis, and inflammatory cell signaling [376]. In neurodegenerative diseases, oxidative stress, energy metabolism disorders, and disease-related protein alterations lead to Ca2^+^-dependent synaptic dysfunction, impaired plasticity, and neuronal death. Dysregulation of the Bcl-2-Ca^2+^ signaling axis has been associated with the progression of AD. Targeting Ca^2+^ checkpoints, such as G protein-coupled receptors, ion channels, Ca^2+^-binding proteins, transcription networks, and ion exchangers, to maintain Ca^2+^ homeostasis may represent novel therapeutic targets [389,390,391]. In addition, the dysregulation of metal metabolism, including iron (Fe) and copper (Cu), also contributes to the regulation of NDDs and the induction of cell death. 

Therefore, we discussed the use of metal chelators related to ferroptosis and cuproptosis [224,225,226,227,228]. Studies have shown that Liproxstatin-1, a specific inhibitor of ferroptosis, can prevent amyloid-beta (Aβ)-induced neuronal death and memory loss [392]. Deferoxamine (DFO), an iron chelator, has demonstrated beneficial effects in clinical trials for AD patients and improves cognitive deficits induced by iron overload in APP/PS1 transgenic mice by inhibiting the processing of amyloid precursor protein (APP) [393,394,395]. Specifically, metal–protein attenuating compounds (MPACs) inhibit Aβ production and toxicity formation by chelating Cu or zinc (Zn) ions, showing efficacy in multiple clinical trials by slowing disease progression and improving cognitive function [396,397,398,399]. Additionally, Fe chelators, such as epigallocatechin-3-gallate (EGCG), and Cu chelators, such as amentoflavone, play significant roles in regulating neuroinflammation and the progression of NDDs [400,401,402].

The pathways of PCD in NDDs represent a complex process regulated by multiple factors. Beyond targeting key elements responsible for executing functions as therapeutic targets, it is crucial to focus on the balance and coordination of multiple pathways. The normal function and balance of autophagy and apoptosis are vital for neuronal homeostasis, and their dysfunction leads to the onset of neurodegenerative diseases. Regulation of some autophagy and apoptosis modulators must be carried out cautiously to avoid excessive autophagy, which can lead to cell death, or excessive inhibition of apoptosis, which can result in the accumulation of toxic substances [403]. In AD, the c-Jun N-terminal kinase (JNK) pathway is upregulated, leading to a reduction in anti-apoptotic protein expression and triggering Janus kinase-signal transducer and activator of transcription (JAK-STAT)/caspase-3 axis-mediated apoptosis. Meanwhile, the phosphoinositide 3-kinase (PI3K)-Akt-mammalian target of rapamycin (mTOR) pathway regulates the imbalance between autophagy and apoptosis, and this balancing act also impacts PD, HD, frontotemporal dementia (FTD), and ALS [214,404,405]. 

Consequently, mTOR inhibitors, such as rapamycin, have shown broad prospects in targeting PCD for the treatment of NDDs [406,407]. Moreover, the upregulation of autophagy can extend the lifespan of HD mice by clearing aggregates containing HTT [408]. Autophagy defects and autophagosome accumulation are attributed to Beclin-1. Beclin-1 serves as a molecular platform for initiating autophagosome formation, and its interaction with anti-apoptotic protein Bcl-2 or inflammasomes leads to autophagy dysfunction and promotes the onset of PD [409,410]. Studies have revealed that the autophagy adaptor protein p62 (sequestosome 1, SQSTM1) mediates the degradation of survival motor neuron (SMN) through interaction, resulting in reduced autophagosome clearance and overactivation of mTOR complex 1 (mTORC1) signaling in spinal muscular atrophy (SMA) neurons [411]. Lowering p62 levels significantly enhances the therapeutic effects for SMA [411].

## 4. Conclusions

Since the discovery of neurodegenerative diseases (NDDs), significant progress has been made in component identification, understanding pathogenesis, development, treatment, and regulation. Current drug development and treatment strategies are based on the exploration and understanding of these mechanisms. As the role of programmed cell death (PCD) in the mechanistic network of NDDs has become clearer, we have gained a deeper understanding of these diseases, creating more possibilities for mechanistic research and clinical treatment. However, breakthroughs based on mechanistic research have been limited, and the translation to clinical progress has encountered bottlenecks. Firstly, in terms of mechanistic exploration, the challenges lie in the insufficient depth of overall research, limited individual research directions and progress, and the inability to integrate multi-spatial, multi-omics, and multi-pathway approaches to establish a comprehensive interpretation. We still need more experimental methods to elucidate the specific roles and dynamic changes of key factors such as the caspase family and B-cell lymphoma 2 (Bcl-2) family in the PCD pathways of diseases and to develop targeted drugs and strategies. Secondly, in terms of clinical development, there is a lack of sufficiently realistic experimental models. Disease models, constrained by research progress, can only replicate relatively singular phenotypes, and there is also a lack of validation of the reliability of existing research findings. Additionally, drug clinicalization faces challenges such as a shortage of development ideas, severe homogenization within the same category, and limitations on drug delivery pathways imposed by the blood–brain barrier. Given these issues, it is crucial to conduct comprehensive assessments of feasibility and biosafety during research to ensure that the final clinical strategies and drugs meet contemporary pharmacokinetic and biosafety standards.

In summary, our review first elucidates the current research on cell death mechanisms by detailing apoptosis, necroptosis, pyroptosis, ferroptosis, cuproptosis, and other forms of PCD. Furthermore, we discuss the roles and processes of various PCD pathways in regulating networks within NDDs, such as Alzheimer’s disease (AD), Parkinson’s disease (PD), Huntington’s disease (HD), amyotrophic lateral sclerosis (ALS), spinal muscular atrophy (SMA), multiple sclerosis (MS), traumatic brain injury (TBI), and stroke. Finally, we briefly introduce disease treatment strategies and approaches targeting key factors in these pathways, including inflammation, oxidative stress, and metabolic balance, thereby establishing a common link between these complex biological processes. Through these three aspects, we preliminarily construct the relationship network between PCD and NDDs, which deepens our understanding of the corresponding mechanisms and identifies potential therapeutic targets for development. Our discussion leads to the following conclusions: (1) The regulation of NDDs by PCD involves the collaborative interaction of multiple factors and pathways. (2) Based on the common understanding of multiple systems, we can overcome the limitations of focusing on single research factors and drug choices, allowing for new experimental attempts. (3) The practical significance of this review lies in providing a reference for the development of therapeutic targets and strategies through elucidation from different perspectives.

As a recognized challenge in human disease, the treatment of NDDs has consistently faced significant obstacles. Based on the current research progress, we are still unable to achieve a complete cure for these diseases, and our ability to prevent their onset remains limited. At best, we can only slow disease progression or alleviate symptoms. Nevertheless, we anticipate that in the near future, the development of drugs targeting PCD for the treatment of NDDs and their clinical application will help improve the condition of patients with neurodegenerative diseases. Despite numerous challenges, we have taken a significant step forward in humanity’s battle against these diseases. We hope to make the greatest possible contribution to curing these diseases within the limits of our current capabilities.

## Figures and Tables

**Figure 1 ijms-25-09947-f001:**
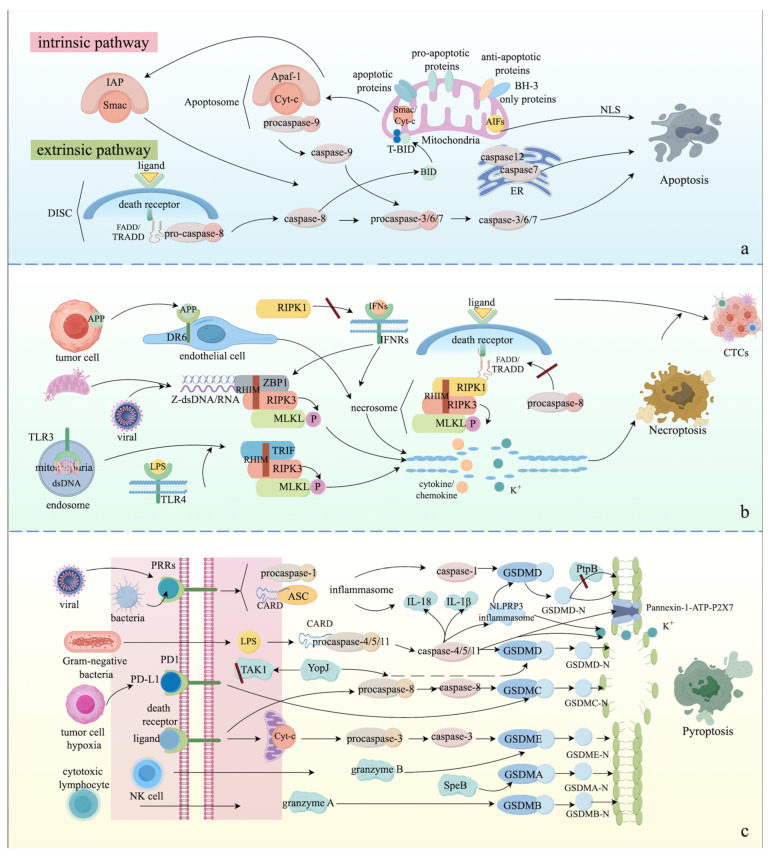
The execution mechanisms of apoptosis, necroptosis, and pyroptosis are described in detail. (**a**) Both the intrinsic and extrinsic pathways of apoptosis are illustrated, highlighting the key signaling molecules and processes involved; (**b**) the assembly and function of various necroptosome structures are depicted, emphasizing the key proteins and their roles; (**c**) the pathways of pyroptosis execution are presented, detailing the stimuli and their effects on cellular components. The red line in the image signifies obstruction or limited functionality. Please refer to the original text for a detailed description of this content. Abbreviations: AIFs, apoptosis-inducing factors; Apaf-1, apoptosis protease activating factor-1; APP, amyloid precursor protein; ASC, apoptosis-associated speck-like protein containing a CARD; BH3-only proteins, Bcl-2 homology 3 domain only proteins; BID, BH3-interacting-domain death agonist; CARD, caspase recruitment domain; CTCs, circulating tumor cells; Cyt-c, cytochrome-c; DISC, death inducing signaling complex; DR6, death receptor 6; dsDNA, double-stranded DNA; ER, endoplasmic reticulum; FADD, Fas-associated death domain; GSDM, gasdermin; IAPs, inhibitors of apoptosis proteins; IFNARs, interferon alpha receptors; IFNs, interferons; IL-18, interleukin-18; IL-1β, interleukin-1β; K, potassium; LPS, lipopolysaccharide; MLKL, mixed-lineage kinase-like; NK, natural killer; NLR, nucleotide-binding oligomerization domain-like receptor; NLRP3, NLR family pyrin domain containing 3; NLS, nuclear localization signal; PD-1, programmed death 1; PD-L1, programmed cell death-ligand 1; PRRs, pattern recognition receptors; PtpB, protein tyrosine phosphatase B; RHIM, RIP (receptor-interacting protein) homology interaction motifs; RIPK1, serine/threonine protein kinase 1; RIPK3, serine/threonine protein kinase 3; Smac, small mitochondria-derived activator of caspase; SpeB, streptococcal pyrogenic exotoxin B; TAK1, TGF-β-activated kinase 1; T-BID, truncated BID; TLR3, toll-like receptor 3; TLR4, toll-like receptor 4; TRADD, TNF receptor-associated death domain; TRIF, TIR-domain-containing adapter-inducing interferon-β; YopJ, yersinia outer protein J; ZBP1, Z-DNA/RNA-binding protein; Z-dsDNA/RNA, Z-form double-stranded DNA/RNA.

**Figure 2 ijms-25-09947-f002:**
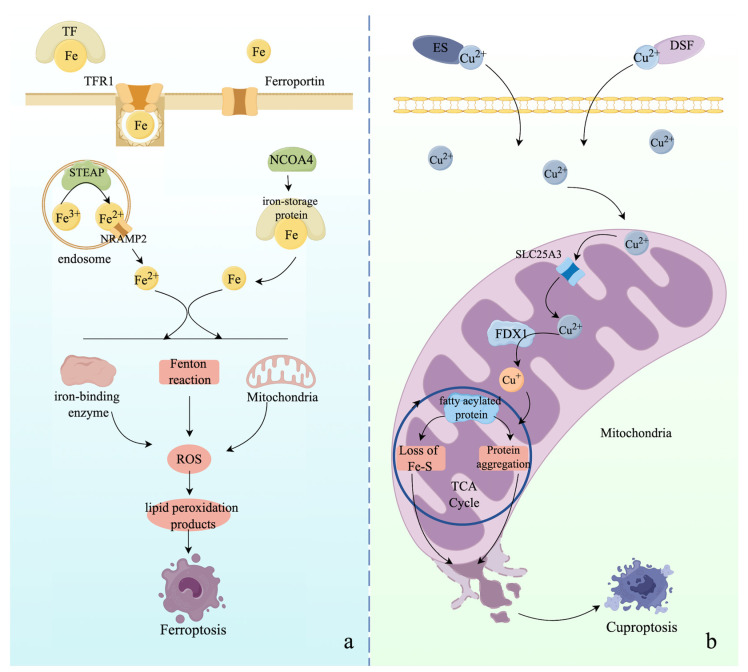
The pathways of cuproptosis and ferroptosis are illustrated. (**a**) Ferroptosis is a form of iron-dependent programmed cell death resulting from intracellular iron overload. The figure depicts the lipid peroxidation induced by dysregulated iron metabolism and the subsequent execution of ferroptosis; (**b**) abnormal copper metabolism and accumulation can lead to protein toxicity, mitochondrial damage, and cuproptosis. For details, refer to the corresponding section of this article. Abbreviations: Cu, copper; DSF, disulfiram; ES, elesclomol; FDX1, ferredoxin 1; Fe, iron; NCOA4, nuclear receptor coactivator 4; NRAMP2 (also known as SLC11A2), natural resistance-associated macrophage protein 2; ROS, reactive oxygen species; S, sulfur; SLC25A3, solute carrier family 25 member 3; STEAP, six-transmembrane epithelial antigen of prostate; TCA, tricarboxylic acid cycle; TF, transferrin; TFR1, transferrin receptor 1.

**Figure 3 ijms-25-09947-f003:**
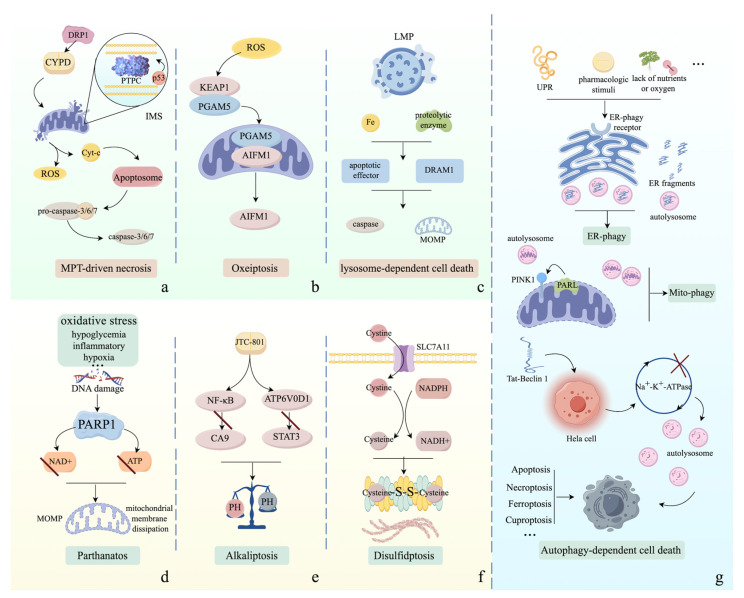
An overview of the mechanisms of various other forms of PCD. (**a**) MPT-driven necrosis is mediated by the activation of CYPD and the formation of PTPC, leading to a loss of selective permeability of the inner mitochondrial membrane, resulting in cell swelling and membrane rupture; (**b**) oxeiptosis is a form of cell death induced by oxygen radicals and mediated by the hyperactivation of the KEAP1-PGAM5-AIFM1 signaling cascade; (**c**) LDCD is a form of cell death caused by changes in lysosomal membrane permeability, resulting in the leakage of lysosomal contents and subsequent alterations in mitochondrial outer membrane permeability; (**d**) parthanatos is a form of cell death induced by DNA damage, resulting in the overactivation of PARP1; (**e**) alkaliptosis is a form of cell death induced by intracellular alkalinization caused by JTC-801, an opioid receptor-like 1 (ORL1) receptor selective antagonist [5]; (**f**) in cells with high SLC7A11 expression, increased cystine uptake leads to NADPH depletion, abnormal disulfide bond formation, cytoskeletal collapse, and disulfidptosis; (**g**) aberrant autophagy leads to excessive ER-phagy, excessive mitophagy, and ADCD. For details, refer to the corresponding section of this article. The red line in the image signifies obstruction or limited functionality. Please refer to the original text for a detailed description of this content. Abbreviations: ADCD, autophagy-dependent cell death; AIFM1, apoptosis-inducing factor mitochondria-associated 1; ATP, adenosine triphosphate; ATP6V0D1, ATPase H+ transporting V0 subunit d1; ATPase, adenosine triphosphatase; CA9, carbonic anhydrase 9; CYPD, cyclophilin D; Cyt-c, cytochrome c; DRAM1, DNA damage-regulated autophagy modulator 1; DRP1, dynamin-related protein 1; ER, endoplasmic reticulum; Fe, iron; IMS, intermembrane space; K, potassium; KEAP1, kelch-like ECH-associated protein 1; LDCD, lysosome-dependent cell death; LMP, lysosomal membrane permeabilization; Mito, mitochondria; MOMP, mitochondrial outer membrane permeabilization; MPT, mitochondrial permeability transition; Na, sodium; NADH, nicotinamide adenine dinucleotide; NADPH, nicotinamide adenine dinucleotide phosphate; NF-κB, nuclear factor κB; PARP1, poly(ADP-ribose) polymerase 1; PGAM, phosphoglycerate mutase; PGAM5, PGAM family member 5; pH, potential of hydrogen; reticulophagy, selective autophagy of the endoplasmic reticulum; PINK1, PTEN-induced kinase 1; PTPC, permeability transition pore complex; ROS, reactive oxygen species; S, sulfur; SLC7A11, solute carrier family 7 member 11; STAT3, signal transducer and activator of transcription 3; UPR, unfolded protein response.

**Figure 4 ijms-25-09947-f004:**
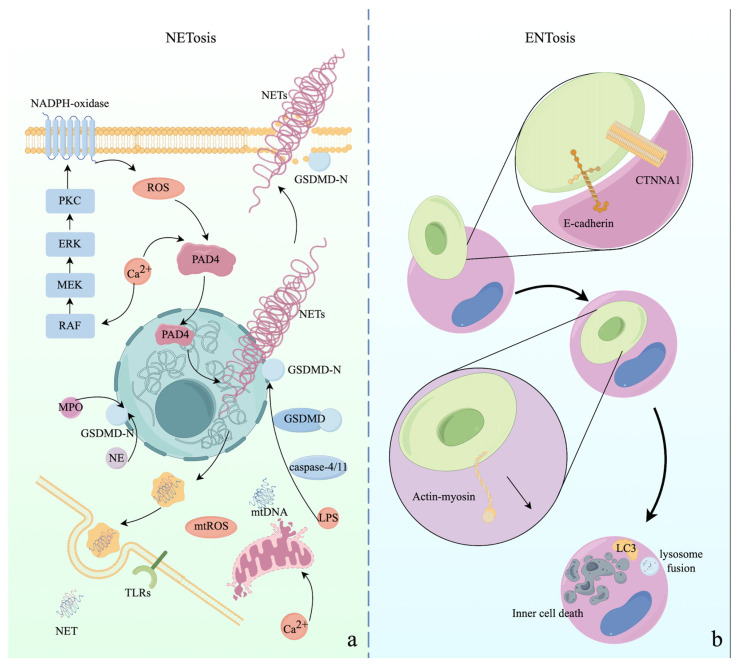
The pathways of NETosis and Entosis are depicted. (**a**) Cellular stress responses induce autophagy, granzyme release and translocation, chromatin decondensation, and cell membrane pore formation, leading to the release of web-like DNA–protein structures and resulting in NETosis; (**b**) cells undergo entosis, an intracellular cell death process, by inserting themselves into neighboring cells through adhesion proteins. For details, refer to the corresponding section of this article. Abbreviations: Ca, calcium; CTNNA1, catenin alpha 1; ENTosis, entotic cell death; ERK, extracellular signal-regulated kinase; GSDM, gasdermin; GSDMD-N,Gasdermin D N-terminal; LC3, microtubule-associated protein 1 light chain 3; LPS, lipopolysaccharide; MEK, MAP kinase kinase; MPO, myeloperoxidase; mtDNA, mitochondrial DNA; mtROS, mitochondrial reactive oxygen species; NADPH, nicotinamide adenine dinucleotide phosphate; NE, neutrophil elastase; NETosis, neutrophil extracellular trap cell death; NETs, neutrophil extracellular traps; PAD4, peptidylarginine deiminase 4; PKC, protein kinase C; RAF, RAF proto-oncogene serine/threonine-protein kinase; TLR, toll-like receptor.

**Figure 5 ijms-25-09947-f005:**
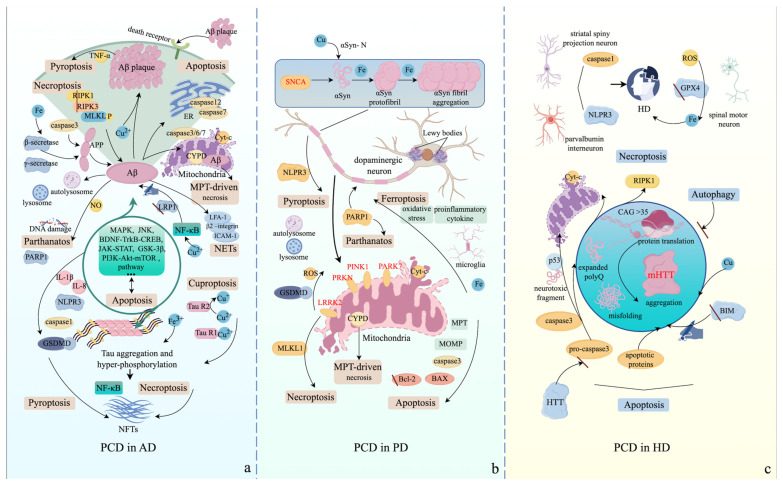
The process of programmed cell death in the development of AD, PD and HD is significant. (**a**) Various forms of programmed cell death play roles in the pathogenesis and progression of AD, including their effects on Tau protein hyperphosphorylation, Aβ plaque formation, and neuronal cell death; (**b**) in PD, various forms of PCD induce dopaminergic neuronal loss and death by promoting αSyn aggregation, leading to mitochondrial dysfunction and neuroinflammation. (**c**) In HD, mHTT induces mitochondrial dysfunction and neuroinflammation by promoting the expression of pro-apoptotic factors and activating necroptosis and ferroptosis. Additionally, the aggregation of mHTT proteins is associated with impaired autophagy, further exacerbating neuronal damage. For details, refer to the corresponding section of this article. The red line in the image signifies obstruction or limited functionality. Please refer to the original text for a detailed description of this content. Abbreviations: AD, Alzheimer’s disease; Akt, protein kinase B; APP, amyloid precursor protein; Aβ, amyloid β; BAX, Bcl-2 associated x-protein; Bcl-2, B-cell lymphoma-2; BDNF, brain-derived neurotrophic factor; BIM, Bcl-2 interacting mediator of cell death; CGA, cytosine-guanine-adenine triplet; CREB, cAMP-response element binding protein; Cu, cuprum; CYPD, cyclophilin D; Cyt-c, cytochrome-c; ER, endoplasmic reticulum; Fe, ferrum; GPX4, glutathione peroxidase 4; GSDMD, gasdermin-D; GSK-3β, glycogen synthase kinase 3β; HD, Huntington’s disease; ICAM-1, intercellular adhesion molecule-1; IL-18, interleukin-18; IL-1β, interleukin-1β; JAK, janus kinase; JNK, c-Jun N-terminal kinase; LFA-1, lymphocyte function-associated antigen 1; LRP1, low-density lipoprotein receptor-related protein 1; LRRK2, leucine-rich repeat kinase 2; MAPK, mitogen-activated protein kinase; mHTT, mutant huntingtin; MLKL, mixed lineage kinase domain-like protein; MOMP, mitochondrial outer membrane permeabilization; MPT, mitochondrial permeability transition; mTOR, mammalian target of rapamycin; NET, neutrophil extracellular traps; NFTs, neurofibrillary tangles; NF-κB, nuclear factor κB; NLRP3, NLR family pyrin domain containing 3; NLR, nucleotide-binding oligomerization domain-like receptor; NO, nitric oxide; PARK7, parkinsonism associated deglycase; PARP1, poly(ADP-ribose) polymerase 1; PCD, programmed cell death; PD, Parkinson’s disease; PI3K, phosphoinositide 3-kinase; PINK1, PTEN induced kinase 1; polyQ, polyglutamine; PRKN, parkin RBR E3 ubiquitin protein ligase; RIPK1, receptor-interacting serine/threonine-protein kinase 1; RIPK3, receptor-interacting serine/threonine-protein kinase 3; ROS, reactive oxygen species; SNCA, alpha-synuclein; αSyn, α-synuclein; STAT, signal transducer and activator of transcription; Tau, microtubule-associated protein Tau; TNF-α, tumor necrosis factor-α; TrkB, tropomyosin receptor kinase B.

**Figure 6 ijms-25-09947-f006:**
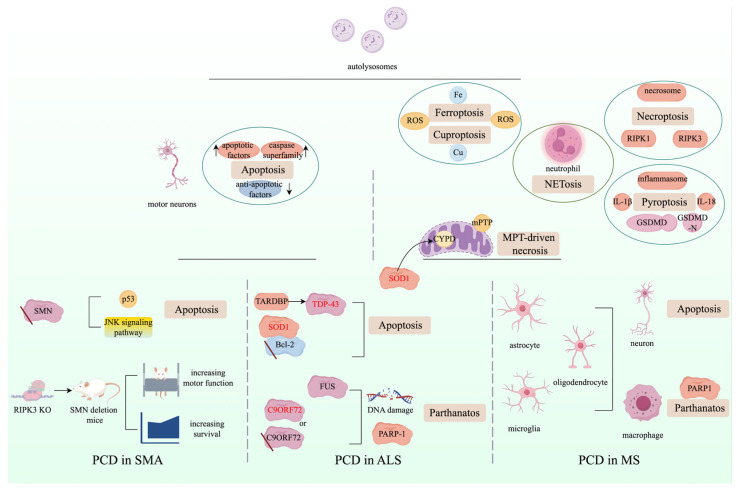
The role of PCD in the progression of ALS, SMA and MS is critical. The figure illustrates the involvement of PCD pathways in the pathogenesis of ALS, SMA, and MS, highlighting both the factors that contribute to disease progression and those that are beneficial for disease control. Additionally, it describes the common pathways through which PCD exerts its effects across these diseases. For more details, refer to the corresponding section of this article. Abbreviations: ADCD, autophagy-dependent cell death; ALS, amyotrophic lateral sclerosis; Bcl-2, B-cell lymphoma-2; C9ORF72, chromosome 9 open reading frame 72; Cu, cuprum; CYPD, cyclophilin D; Fe, ferrum; FUS, fused in sarcoma/translocated in liposarcoma; GSDMD, gasdermin-D; IL-18, interleukin-18; IL-1β, interleukin-1β; JNK, c-Jun N-terminal kinase; KO, knockout; MPT, mitochondrial permeability transition; mPTP, mitochondrial permeability transition pore; MS, multiple sclerosis; NET, neutrophil extracellular traps; NETosis, neutrophil extracellular trap cell death; PARP1, poly(ADP-ribose) polymerase 1; PCD, programmed cell death; RIPK1, receptor-interacting serine/threonine-protein kinase 1; RIPK3, receptor-interacting serine/threonine-protein kinase 3; ROS, reactive oxygen species; SMA, spinal muscular atrophy; SMN, survival motor neuron; SOD1, superoxide dismutase 1; TARDBP, TAR DNA-binding protein; TDP-43, TAR DNA-binding protein 43.

**Figure 7 ijms-25-09947-f007:**
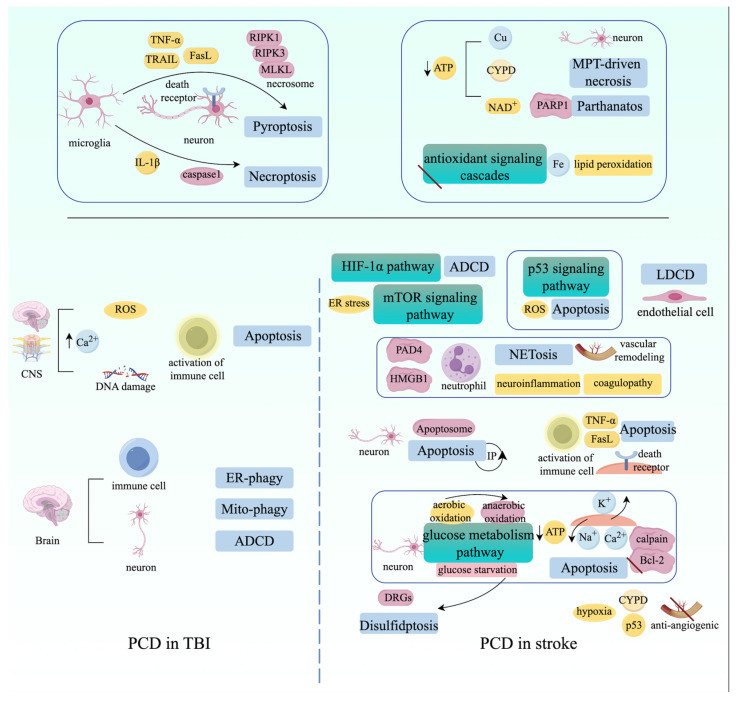
The role of programmed cell death (PCD) in the progression of traumatic brain injury (TBI) and stroke. The figure illustrates various aspects that contribute to disease progression, revealing the role of PCD in these conditions and considerations beneficial for disease control and treatment. Additionally, it describes the pathways through which PCD exerts its effects both individually and collectively in TBI and stroke. For details, refer to the corresponding section of this article. The upward and downward arrows represent an increase and decrease in content or concentration, respectively. The circular arrow signifies that "apoptosis in neurons of the ischemic penumbra may be recoverable." The red line in the image signifies obstruction or limited functionality. Please refer to the original text for a detailed description of this content. Abbreviations: ADCD, autophagy-dependent cell death; ATP, adenosine triphosphate; Bcl-2, B-cell lymphoma-2; Ca, calcium; CNS, central nervous system; Cu, cuprum; CYPD, cyclophilin D; DRGs, dorsal root ganglions; ER, endoplasmic reticulum; FasL, Fas ligand; Fe, ferrum; HIF-1α, hypoxia-inducible factor 1α; HMGB1, high-mobility group box 1; IL-1β, interleukin-1β; IP, ischemic penumbra; K, kalium; LDCD, lysosome-dependent cell death; Mito, mitochondria; MLKL, mixed lineage kinase domain-like protein; MPT, mitochondrial permeability transition; Na, natrium; NAD, nicotinamide adenine dinucleotide; NET, neutrophil extracellular traps; PAD4, peptidylarginine deiminase 4; PARP1, poly(ADP-ribose) polymerase 1; PCD, programmed cell death; RIPK1, receptor-interacting serine/threonine-protein kinase 1; RIPK3, receptor-interacting serine/threonine-protein kinase 3; ROS, reactive oxygen species; TBI, traumatic brain injury; TNF-α, tumor necrosis factor-α; TRAIL, tumor necrosis factor-related apoptosis-inducing ligand.

**Table 1 ijms-25-09947-t001:** Key regulators involved in the iron metabolism and ferroptosis.

Modulators	Functions	Refs.
hepcidin	degrading the ferroportin via ubiquitination	[102]
IRP1/2	promoting the expression of TFR1	[103]
PCBP1	delivering ferrous iron to ferritin	[104]
SLC25A37/8	promoting the absorption of iron	[105]
HO1	catalyzing the synthesis of ferrous iron	[106]
SLC40A1	assisting the export of iron	[107]
SLC39A14	assisting the import of iron	[108]
SLC25A28	regulating the generation of ROS	[109]
SFXN1	regulating the generation of ROS	[110]
PROM2	regulating the storage of iron in ferritin	[111]
PHKG2	modulating the oxidative reactions	[112]
HMOX1	participating in the synthesis of ferrous iron	[113]
SLC11A2	assisting the absorption of iron	[87]
CP	converting ferrous iron to ferric iron	[114]
CISD1/2	participating in the absorption of iron	[115,116]
DMT1	controlling the absorption of iron	[117]
FBXL5	degrading the IRP1 via ubiquitination	[118]
HSF1	regulating the iron metabolism-related genes	[119]
HSPA5	binding to GPX4 to prevent its degradation	[120]
NRF2	inducing the expression of antioxidant genes	[121]
CISD1	alleviating the accumulation of lipid	[122]
ALOXs	facilitating the lipoxygenase	[123]
PEBP1	enhancing the lipid death pathway via 15-LO	[124]
NOXs	facilitating the generation of ROS	[125]
DPP4/CD26	causing the lipid peroxidation	[126,127]
VDAC2/3	activating ferroptotic agonist erastin	[128]
MUC1	activating GPX4	[129]
GCLC	accelerating the synthesis of GSH	[130]
GLS2	increasing the ROS production via αKG	[131]
CARS	inhibiting the generation of GSH	[132]
CHAC1	promoting oxidative reactions	[133]
LSH	promoting SLC7A11 transcription	[134]
FADS2	desaturating the free fatty acids	[135]
ACSL3	upregulating the lipid droplet biogenesis	[136]
ACSL4	shaping cellular composition	[137]
LPCAT3	upregulating polyunsaturated free fatty acid	[138]
PHGDH	upregulating the expression of PHGDH	[139]
G6PD	preventing oxidative reactions via inhibiting POR	[140]
ME1	facilitating the generation of GSH	[141]
PHKG2	regulating lipoxygenase enzyme ALOX5	[142]
HMGCR	increasing GPX4 and CoQ_10_ biosynthesis	[143]
SQLE	preventing oxidative via squalene	[144]
NRF2	regulating the antioxidant-related genes expression	[145]
P53	modulating GPX4 pathway and ROS production	[146]
HIF-1α	improving the expression of SLC7A11 via PMAN	[147]
BACH1	enhancing iron metabolisom-related gene expression	[148]
STAT3	enhancing the expression of GPX4 and SLC7A11	[149]
ATF3	restraining the activity of system Xc^−^	[150]
ATF4	facilitating the expression of SLC7A11	[151]
CHOP	facilitating the expression of CHAC1	[152]
YAP/TAZ	inducing the expression of SLC7A11	[153]

Abbreviation: IRP1/IRP2, iron regulatory protein 1/2; PCBP1, poly(rC)-binding protein 1; SLC25A37/SLC25A38, mitoferrin 1/2; HMOX1, heme oxygenase 1; SLC40A1, solute carrier family 40 member 1; SLC39A14, solute carrier family 39 member 14; SLC25A28, mitoferrin 2; SFXN1, sideroflexin 1; PROM2, prominin 2; PHKG2, phosphorylase b kinase γ-catalytic chain, liver/testis isoform; SLC11A2, natural resistance-associated macrophage protein 2; CP, ceruloplasmin; CISD1/CISD2, CDGSH iron–sulfur domain-containing protein 1/2; ACO1, cytoplasmic aconitate hydratase; DMT1, divalent metal transporter 1; FBXL5, F-box and leucine-rich repeat protein 5; HSF1, heat shock factor 1; HSPA5, heat shock 70-kDa protein 5; NRF2, nuclear factor erythroid 2-related factor 2; ALOXs, arachidonic acid lipoxygenases; PEBP1, phosphatidylethanolamine (PE)-binding protein 1; NOXs, NADPH oxidases; DPP4/CD26, dipeptidyl peptidase-4; VDAC2/3, voltage-dependent anion channel 2/3; MUC1, mucin 1; GCLC, glutamate-cysteine ligase catalytic subunit; GLS2, glutaminase 2; CARS, cysteinyl-tRNA synthetase; CHAC1, cation transport regulator-like protein 1; LSH, lymphoid-specific helicase; SCD, stearoyl-CoA desaturase; FADS2, fatty acid desaturase 2; ACSL3/4, acyl-CoA synthetase long-chain family member 3/4; LPCAT3, lysophosphatidylcholine acyltransferase 3; PHGDH, phosphoglycerate dehydrogenase; G6PD, glucose 6-phosphate dehydrogenase; ME1, malic enzyme 1; αKG, α-ketoglutarate; POR, cytochrome P450 oxidoreductase; ALOX5, arachidonate 5-lipoxygenase; HMGCR, 3-hydroxy-3-methylglutaryl-CoA reductase; SQLE, squalene epoxidase; HIF-1α, hypoxia-inducible factor 1α; BACH1, BTB domain and CNC homolog 1; STAT3, signal transducer and activator of transcription 3; ATF3/4, activating transcription factor 3/4; CHOP, C/EBP homologous protein; YAP/TAZ, Yes-associated protein/Transcriptional coactivator with PDZ-binding motif.

**Table 2 ijms-25-09947-t002:** Hallmark features of aforementioned PCD modalities.

Forms	Immune Features	Morphological Features	Major Inhibitors	Refs.
Apoptosis	TCD	Apoptotic bodies formation; Nuclear condensation; Plasma membrane blebbing; Cell shrinkage.	Z-VAD-FMK; Q-VD-OPh; Z-VAD (OH)-FMK.	[4] [17]
Necroptosis	ICD	Cell swelling and oncosis Rupture of plasma membrane; Swelling of organelles Chromatin condensation.	Nec-1; GSK872; NSA; HS-1371.	[46] [49]
Pyroptosis	ICD	Lack of cell swelling; Rupture of plasma membrane; Cell bubbling; Chromatin condensation.	Ac-YVAD-cmk; VX765; Ac-FLTD-CMK.	[67] [204]
Ferroptosis	ICD	Smaller mitochondria; Rupture of mitochondrial membrane; Decreased mitochondrial cristae; Normal nucleus.	Deferiprone; Ferrostatin-1; Alogliptin; Selenium; CoQ10; Vildagliptin; Vitamin E.	[80] [86] [205]
Cuproptosis	ICD	mitochondrial condensation; Rupture of plasma membrane; ER damage; Chromatin condensation.	NSC689534 EMeramide Penicillamine AT-VI	[7] [154]
MPT-driven necrosis	ICD	Plasma membrane rupture; Swelling of organelles; Lack of inter-nucleosomal DNA fragmentation; Depletion of ATP.	SfA	[5]
ADCD	ICD	Autophagic vacuolization.	Chloroquline; Spactin-1.	[168]
LDCD	ICD	Rupture of lysosome and plasma membrane.	NAC; CA-074Me	[1]
Parthanatos	ICD	Chromatin condensation; Large DNA fragmentation; Loss of cell swelling.	BYK204165; AG-14361; Iniparib.	[184] [185]
Oxeiptosis	TCD	Apoptosis-like morphology.	NAC.	[188]
Alkaliptosis	ICD	Necrosis-like morphology.	NAC; CAY10657; SC514.	[5]
Disulfidptosis	ICD	Cell shrinkage; Nuclear condensation; Formation of aberrant disulfide bonds between actin cytoskeleton proteins; Chromatin condensation.	GLUT inhibitor	[6] [189]
NETosis	TCD or ICD	Rupture of plasma membrane and nuclear membrane; Release of chromatin fragments.	lactoferrin; DNase; Cl-amidine.	[190] [206]
ENTosis	TCD or ICD	Formation of cell-in-cell structure.	C3-toxin; γ-27632	[196]

Abbreviations: MPT, mitochondrial permeability transition; ADCD, autophagy-dependent cell death; LDCD, lysosome-dependent cell death; NETosis, neutrophil extracellular trap cell death; ENTosis, entotic cell death; ICD, immunogenic cell death; TCD, tolerogenic cell death; ER, endoplasmic reticulum; AT-VI, ammonium tetrathiomolybdate (VI); Nec-1, necrostatin-1; GLUT inhibitor, glucose transporter 1 inhibitor; NSA, necrosulfonamide; SfA, sanglifehrin A.

## Data Availability

No new data were created or analyzed in this study. Data sharing is not applicable to this article.

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
