# Peer review of "Significance of Programmed Cell Death Pathways in Neurodegenerative Diseases"

_ijms, 2024, doi:10.3390/ijms25189947_

Round 1

Reviewer 1 Report

Comments and Suggestions for Authors

Review of manuscript entilted: “Significance of Programmed Cell Death Pathways in Neuro-degenerative Diseases” authored by Dong Guo, Zhihao Liu, Jinglin Zhou, Chongrong Ke and Daliang Li

First of all I want to thank you for opportunity to review this interesting manuscript.

In the presented manuscript, authors review current state of art about programmed cell death pathways in neurodegenerative diseases. Authors undertook huge task and in my opinion put huge effort to address the topic in comprehensive manner. Manuscript is divided into four paragraphs. In the introduction part, authors highlight what will be furtherly discussed among their manuscript. Next paragraph provides details about each programmed cell death pathway, this part is pretty long and enriched with nice figures illustrating each mentioned pathway. In the third paragraph authors described what is essential in their review also supported by appropriate figures. Last paragraph contains brief conclusions. Manuscripts includes impressive number of citations (410).

Overall, manuscript is very comprehensive and not so reader-friendly. When, I was reading this manuscript I felt overwhelmed by many information which are provided here and the amount of abbreviations, which is understandable due to broad topic. I think that manuscript would greatly benefit by polishing it. Despite some flaws (which are listed below) this review is a very comprehensive, detailed summary of current knowledge about programmed cell death in neurodegenerative diseases and will be valuable.

Major concerns:

  • Introduction – I believe that it may be shortened by deletion of unnecessary repetition for example lines 38-43 are repeated in 70-74. I understand the point of this but I think that lines 70-74 can be replaced by “In this review, we provide a comprehensive overview of aforementioned PCD subroutines” or something alike;
  • Across the manuscript we find plenty of abbreviations, what is understandable but some of them are introduced after first usage of the full name, some of them are introduced multiple times (example line 605 “blood-brain barrier”, then in line 882 “blood-brain barrier (BBB)” the abbreviation is provided);
  • I think that polishing is required since the manuscript is really challenging;

Minor concerns:

  • Maybe too detailed description of each signalling pathway in the second paragraph;
  • Minor spellcheck is required;

Reviewer 2 Report

Comments and Suggestions for Authors

In this manuscript Guo et al, extensively review the implications of PCD in neurodegenerative diseases.

 Comments:

1)       Do the authors have publications on this topic? If not, what is the meaning of this review? What are the new ideas/perspectives or hypotheses that the authors contribute in this review? Is it just a compendium of currently existing information?

2)        Abstract. Typically, PCD signaling events are precisely regulated by various biomolecules in both spatial and temporal contexts to promote neuronal development, establish neural architecture, and shape the central nervous system (CNS).

                The role of PCD is not restricted to CNS

3)       Vitamin E should be included as ferroptosis inhibitor.

4)       The information provided in section Therapeutic strategies targeting PCD signaling pathways in NDDs should be summarize in a table, specifying t disease, biological model, intervention pathway, clinical trials, outcome and references.

Round 2

Reviewer 2 Report

Comments and Suggestions for Authors

The authors have addressed all my concerns